# *Pyphe*, a python toolbox for assessing microbial growth and cell viability in high-throughput colony screens

Stephan Kamrad[1,2], María Rodríguez-López[1], Cristina Cotobal[1], Clara Correia-Melo[2], Markus Ralser[2,3]*, Jürg Bähler[1]*

[1]University College London, Institute of Healthy Ageing, Department of Genetics, Evolution and Environment, London, United Kingdom; [2]The Francis Crick Institute, Molecular Biology of Metabolism Laboratory, London, United Kingdom; [3]Charité Universitaetsmedizin Berlin, Department of Biochemistry, Berlin, Germany

**Abstract** Microbial fitness screens are a key technique in functional genomics. We present an all-in-one solution, *pyphe*, for automating and improving data analysis pipelines associated with large-scale fitness screens, including image acquisition and quantification, data normalisation, and statistical analysis. *Pyphe* is versatile and processes fitness data from colony sizes, viability scores from phloxine B staining or colony growth curves, all obtained with inexpensive transilluminating flatbed scanners. We apply *pyphe* to show that the fitness information contained in late endpoint measurements of colony sizes is similar to maximum growth slopes from time series. We phenotype gene-deletion strains of fission yeast in 59,350 individual fitness assays in 70 conditions, revealing that colony size and viability provide complementary, independent information. Viability scores obtained from quantifying the redness of phloxine-stained colonies accurately reflect the fraction of live cells within colonies. *Pyphe* is user-friendly, open-source and fully documented, illustrated by applications to diverse fitness analysis scenarios.

*For correspondence:
markus.ralser@crick.ac.uk (MR);
j.bahler@ucl.ac.uk (JB)

**Competing interests:** The authors declare that no competing interests exist.

## Introduction

Colony fitness screens are a key assay in microbial genetics. The availability of knock-out libraries has revolutionised reverse genetics and enabled the field of functional genomics (*Giaever and Nislow, 2014*). Simultaneously, large collections of wild isolates (*Jeffares et al., 2015*; *Peter et al., 2018*), as well as synthetic populations (*Bloom et al., 2013*; *Cubillos et al., 2013*), have proven a powerful tool to study complex traits. More recently, the systematic measurement of fitness for hundreds of conditions and/or hundreds/thousands of strains in parallel is driving our systems-level understanding of gene function (*Brochado et al., 2018*; *Costanzo et al., 2016*; *Kuzmin et al., 2018*; *Nichols et al., 2011*).

Microbial phenomics screens generally follow a workflow where strains are arranged in high-density arrays (e.g. 384 or 1536 colonies per plate) and transferred using a colony-pinning robot or manual replicator. Image analysis software enables fast and precise quantification of colony sizes and other phenotypes (*Bischof et al., 2016*; *Kritikos et al., 2017*; *Lawless et al., 2010*; *Memarian et al., 2007*; *Wagih and Parts, 2014*). Colony-size data is prone to noise and technical variation between areas on the same plate and across plates and batches, some of which can be corrected by normalisation procedures (*Baryshnikova et al., 2010*; *Blomberg, 2011*; *Zackrisson et al., 2016*). Finally, differential fitness is assessed statistically, for which specialised approaches are available (*Collins et al., 2010*; *Collins et al., 2006*; *Wagih and Parts, 2015*).

Most screens use a single image or timepoint per plate (an endpoint measurement). Potentially more information is contained in the growth of colony sizes over time and a low-resolution time

course of colony sizes can be used to fit growth models to population size data (*Addinall et al., 2011*; *Banks et al., 2012*; *Shah et al., 2007*). High-resolution image time series contain potentially even more information and have been used to determine lag phases (*Levin-Reisman et al., 2014*). Recently, highly precise fitness determination has been achieved by high-resolution, transilluminating time course imaging and growth curve analysis (*Takeuchi et al., 2014*) and combined with a reference grid normalisation (*Zackrisson et al., 2016*). The parallel use of commercially available scanners, combined with high-density arrays of colonies can enable growth curve-based phenotyping at very large scales, but poses challenges in terms of data storage, processing, equipment and the need for temperature-controlled space.

The dead-cell stain phloxine B can provide an additional phenotypic readout related to the proportion of dead cells in a colony. Phloxine B has been used to assess the viability of cells in budding yeast by microscopy (*Tsukada and Ohsumi, 1993*) and in fission yeast colonies (*Matynia et al., 1998*). When applied in a screening context, colonies are assigned a score which reflects the 'redness' of the colony to serve as an additional quantitative phenotype that can be used for downstream analysis (*Lie et al., 2018*).

Despite the popularity and importance of microbial colony screens, a consensus data framework has so far not emerged. In our laboratories, fitness screens are an essential technique used on a variety of scales, from a handful of plates to several thousand, and by researchers with varying bioinformatics skills. To enable and standardise data analysis workflows, we have developed a bioinformatics toolbox with a focus on being versatile, modular and user friendly. *Pyphe* (*py*thon package for *phe*notype analysis) consists of 6 command-line tools, each performing a different workflow step as well as the underlying functions, provided as a python package to expert users.

We illustrate the use of *pyphe* by investigating the growth dynamics of 57 natural *S. pombe* isolates. We show that the spatial correction implemented in *pyphe*, based on that proposed by *Zackrisson et al., 2016*, is effective in reducing measurement noise without overcorrection. Late endpoint measurements are shown to provide similar readouts to maximum slopes, but with lower precision. We then investigate the relationship between colony sizes and viability scores in a broad panel of *S. pombe* knock-out strains in over 40 conditions and find that the two approaches provide orthogonal and independent information. Using imaging flow cytometry, we link colony redness scores to the percentage of dead cells in a colony and show that phloxine B staining provides similar results as a different live/dead stain.

## Results

### *Pyphe* enables analysis pipelines for fitness-screen data

The *pyphe* pipeline is designed to take different fitness proxies as input: endpoint colony sizes, colony growth curves or endpoint colony viability estimates from phloxine B staining (*Figure 1*). Image acquisition, image analysis, growth-curve analysis, data normalisation and statistical analysis are split into separate tools which can be assembled into a pipeline as required for each experiment and combined with other published tools, e.g. *gitter* (*Wagih and Parts, 2014*) for image quantification. Each tool takes and produces human-readable data in text/table format.

In a typical workflow, images are acquired using *pyphe-scan* which provides an interface for image acquisition using SANE (Scanner Access Now Easy) on a Linux-type operating system. It handles plate numbering, cropping and flopping, and format conversion functionality for large image stacks. Optionally, image time-series can be recorded. *Pyphe-scan* was written to work with EPSON V800 scanners, the newer model in the series previously used by others (*Takeuchi et al., 2014*; *Zackrisson et al., 2016*).

Colony properties are then quantified from images using *pyphe-quantify* which can operate in three different modes. In *batch* mode (for colony-size quantification using grayscale transmission scanning) or *redness* mode (colony-viability estimation using phloxine B and reflective colour scanning), it separately analyses all images that match the input pattern (by default all jpg images in working directory), producing a csv table and qc image for each. In *timecourse* mode, colony positions are determined in the last image and the mask is applied to all previous images, extracting background-subtracted sums of pixel intensities for each colony/spot and producing a single table with the growth curves (one per column). *Pyphe quantify* reports a wide range of colony properties:

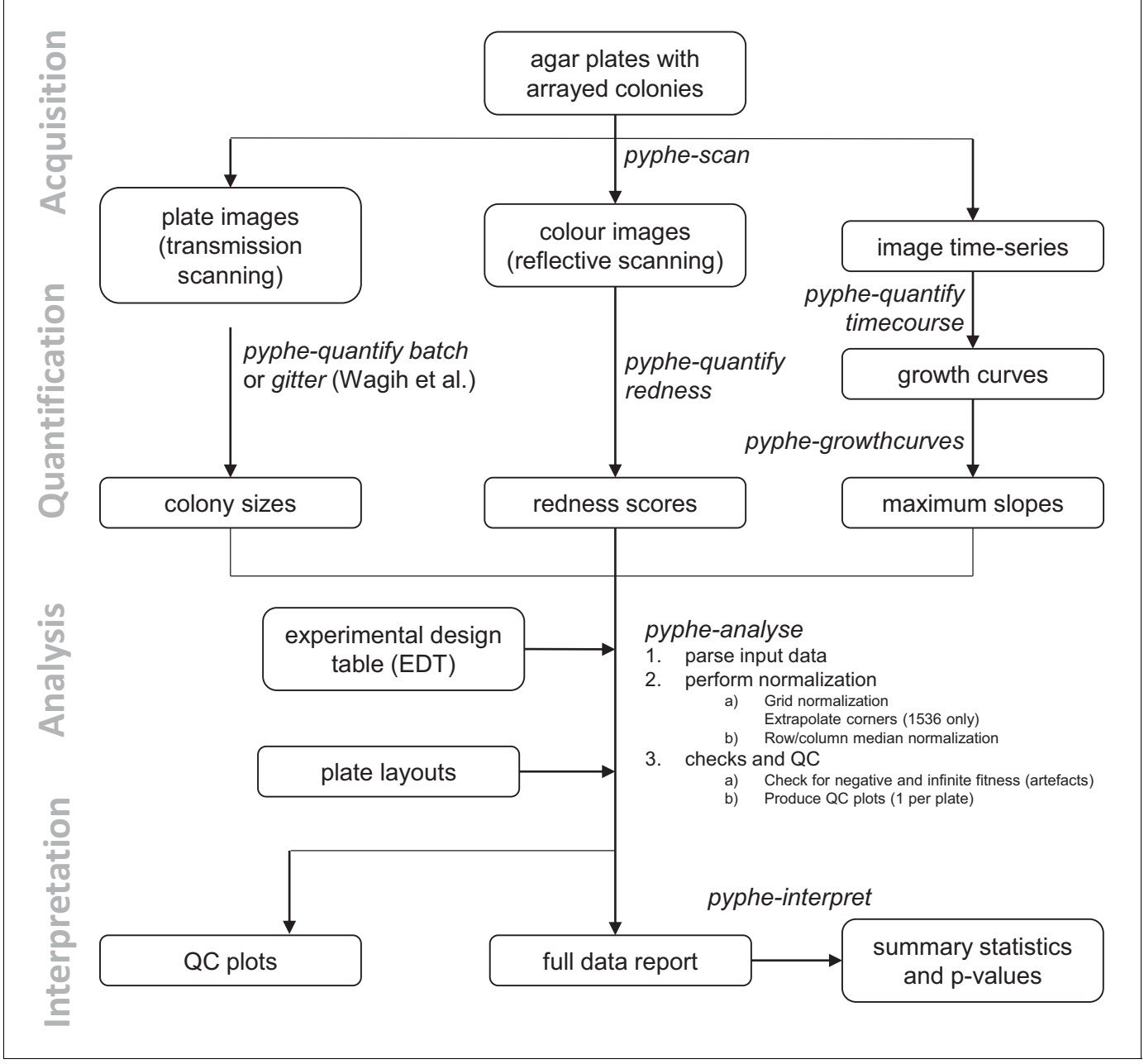

**Figure 1.** Data processing workflows using *pyphe*. *Pyphe* is flexible and can use several fitness proxies as input. In a typical endpoint experiment, plate images are acquired using transmission scanning and colony sizes are extracted using *pyphe-quantify* or the R package *gitter* (***Wagih and Parts, 2014***). Alternatively or additionally, plates containing phloxine B are scanned using reflective scanning and analysed with *pyphe-quantify* in redness mode to obtain redness scores reflecting colony viability. Alternatively, image time series can be analysed with *pyphe-quantify* in *timecourse* mode and growth curve characteristics extracted with *pyphe-growthcurves*. *Pyphe-analyse* analyses and organises data for collections of plates. It requires an Experimental Design Table (EDT) containing a single line per plate and the path to the data file, optionally the path to the layout file, and any additional metadata the user wishes to include. Data is then loaded and the chosen normalisation procedures are performed. QC plots are produced and the entire experiment data is summarised in a single long table. This table is used by *pyphe-interpret* which produces a table of summary statistics and p-values for differential fitness analysis.

The online version of this article includes the following figure supplement(s) for figure 1:

**Figure supplement 1.** Image analysis with *pyphe-quantify*, described in Appendix 1.

**Figure supplement 2.** Spatial normalisation with *pyphe-analyse*, described in Appendix 2.

colony area, overall intensity (an estimator that reflects thickness as well as area), circularity, perimeter and centroid coordinates, making this tool useful in cases where colonies are not arrayed. Image pixel darkness is known to scale non-linearly with true colony thickness/cell number (*Zackrisson et al., 2016*). Fitness estimates reported by *pyphe-analyse* are therefore related but not strictly the same as cell counts. If absolute population sizes are required for an experiment, the Scan-o-matic pipeline offers suitable calibration functionalities (*Zackrisson et al., 2016*). *Pyphe-quantify* algorithms are described in detail in Appendix 1 and *Figure 1—figure supplement 1*.

Spatial normalisation is performed for each plate and data across all plates are aggregated using *pyphe-analyse* to produce a single table for downstream hit calling and further analysis. *Pyphe* implements a grid normalisation procedure based on the one previously described (*Zackrisson et al., 2016*) as well as row/column median normalisation. Both strategies produce relative fitness estimates where a value of 1 corresponds to the fitness of the grid strain or the plate median respectively. We propose an improved placement of the grids in 1536 format (*Figure 1—figure supplement 2*) and implemented checks for missing colonies and normalisation artefacts. The main output is a single long table, containing one row per colony, with all position-, strain-, meta- and fitness-data as well as details about the normalisation. Algorithms are further described in Appendix 2 and *Figure 1—figure supplement 2*.

Finally, differential fitness is assessed using *pyphe-interpret* which produces summary statistics and p-values based on the complete data report from *pyphe-analyse* (Appendix 3). *Pyphe-interpret* gives users the option to either test for differences between strains in the same condition or between the same strain in different conditions. The latter is the recommended option for testing for condition-specific growth effects compared to a control condition.

## Effective normalisation reduces noise and bias in data

*Pyphe* is designed to use different fitness proxies as input. In particular, it can use either maximum growth rates extracted from growth curves or endpoint colony size measurements. Previous studies have reported that information from growth curves are more precise (*Zackrisson et al., 2016*), but their acquisition requires substantially higher investment and produces large amounts of image data. While lower precision could be easily compensated by a higher number of replicates, growth curves provide the additional advantage that they capture the entire growth phase instead of a static snapshot. The results obtained in endpoint measurements might, therefore, depend on the timepoint used for the measurement. For example, the fitness of a strain with a long lag phase but high maximum growth rate may be underestimated if an early timepoint is chosen.

To assess the extent to which the choice of the timepoint matters, we recorded image time series for 57 *S. pombe* wild strains growing in 1536 spots per plate in approximately 20 replicates on 8 different media (*Supplementary file 1*). The conditions were designed to produce different growth rates and dynamics, and included mixes of different carbon sources with yeast extract as nitrogen source in rich media and different nitrogen sources in minimal media. These strains are genotypically and phenotypically diverse and display a broad range of growth characteristics (*Jeffares et al., 2015*). First, colony areas were extracted with *gitter* (*Wagih and Parts, 2014*), and relative, corrected colony sizes were computed for each image using the grid normalisation implemented in *pyphe-analyse* and averaged for each strain. We show an exemplary analysis of a single condition (standard rich media) in *Figure 2* and a detailed analysis of all conditions in *Figure 2—figure supplement 1* and *2*. Relative colony sizes remained largely constant after the period of fast growth had come to end at roughly 16 hr (*Figure 2A*). Concordantly, a correlation matrix of all timepoints showed near perfect correlation of timepoints with the 48 hr end point from 16 hr (*Figure 2Biii*). Notably, all timepoints were correlated with the initial timepoint, albeit much lower, suggesting a significant bias introduced by the amount of initially deposited biomass. In our hands, this problem is more pronounced with wild strains than with knock-out collections as the former exhibit a variable degree of cell aggregation. However, we overcome this issue by reporting strain fitness as a ratio of growth in an assay condition relative to a control condition, in which case this bias is neutralised. We next analysed image timeseries with *pyphe-quantify* in timecourse mode, extracted slopes with *pyphe-growthcurves* (Appendix 4) and applied grid correction in *pyphe-analyse*. Later timepoints generally showed a much better correlation with maximum growth rate compared to early ones or those taken when growth is most rapid (*Figure 2* Bi+ii). Across all conditions, the median correlation of corrected maximum slopes with corrected colony sizes at the final timepoint was 0.95 (*Figure 2—*

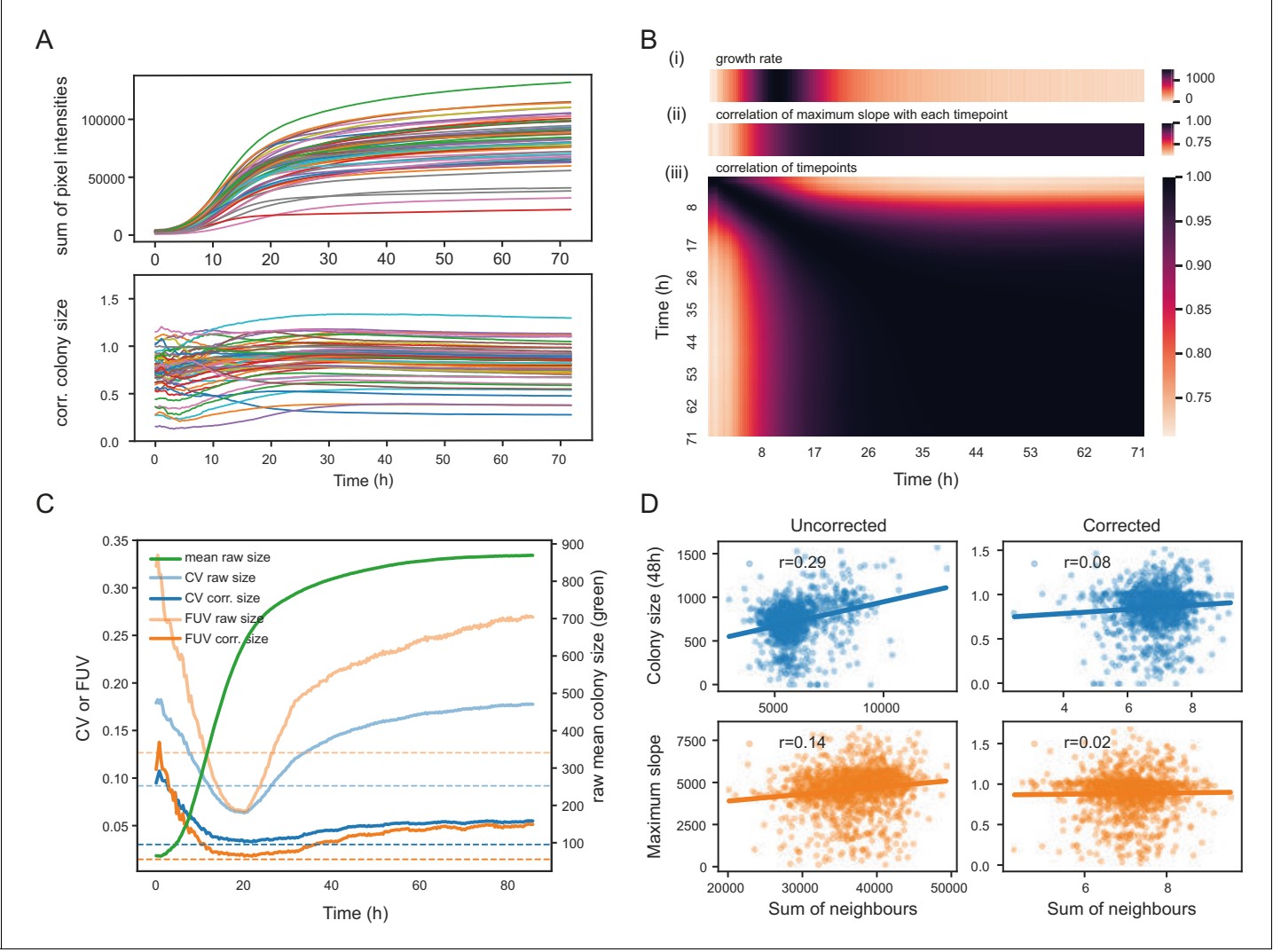

**Figure 2.** Normalisation strategies for growth curves and endpoints. (**A**) Growth curves of 57 wild *S. pombe* strains (average of approximately 20 replicates each) before (top) and after (bottom) correction. Corrected colony sizes describe the fitness relative to the standard laboratory strain (*972*) after grid correction. (**B**) Late endpoint measurements are tightly correlated with maximum slopes. (i) Average growth rates (mean difference in sum of pixel intensities between consecutive timepoints) across all strains. (ii) Pearson correlation of each individually corrected timepoint with corrected maximum slope of growth curves. The correlation increases throughout the rapid growth curve and then maintains high levels as the phase of fast growth comes to an end. (iii) Pearson correlation matrix of all corrected timepoints (averaged by strain prior to correlation analysis). (**C**) Coefficient of variation (CV, blue) and fraction of unexplained variance (FUV, orange) for corrected and uncorrected colony sizes throughout the growth curve. Dashed lines are the same values computed based on maximum slopes. The average growth curve of the control strain is shown in green (based on colony sizes extracted with *gitter*). The normalisation procedure maintains noise at low levels even in later growth. Endpoint measurements contain slightly more noise than slope measurements. (**D**) Scatter plots of colony fitness estimates dependent on the sum of colony fitness of its 8 neighbours. A positive correlation, such as seen for the uncorrected readouts, points to spatial biases within plates (specific regions of a plate growing slower/faster, for example due to temperature, moisture or nutrient gradients). A negative correlation would be expected for competition effects. Without correction, regional plate effects dominate over competition effects and these are efficiently removed during grid correction. Importantly, the correction does not result in a negative correlation, a potential side-effect of correcting colony sizes by comparing it to the size of neighbouring controls, which would lead to phenotypes becoming artificially more extreme.

The online version of this article includes the following figure supplement(s) for figure 2:

**Figure supplement 1.** 57 wild strains on different carbon and nitrogen sources.

**Figure supplement 2.** 57 wild strains on different carbon and nitrogen sources (cont'd).

**Figure supplement 3.** 57 wild strains analysis summary.

*figure supplement 3*). We conclude that late timepoints should be chosen for endpoint measurements, when the readout is stable and correlates well with the maximum growth rate.

The choice of timepoint also affects the level of noise. For uncorrected colony sizes, the coefficient of variation (CV, ratio of the standard deviation to the mean) of 96 replicates of the control strain, dispersed evenly in the plate, dropped steadily during the rapid growth phase, reaching a minimum around 20 hr when it started to rise again (*Figure 2C*). This is likely due to edge and other spatial effects which affect later growth as nutrients deplete and plates start to dry unevenly. After normalisation, the CV was generally lower, and this later rise in noise could be compensated so that the CV remained near its minimum. The CV of the maximum slopes was lower than obtained with endpoints. However, CV values alone are insufficient to judge the effectiveness of a normalisation strategy, as it reflects precision of the reported values but not the method's ability to delineate differences between strains. As an additional indicator, we therefore used the ratio of the variance of the controls and the variance of the entire dataset, the fraction of unexplained variance (FUV), which indicates the level of noise relative to the biological signal in the data. Overall, the FUV behaved similarly to the CV and was at a minimum at around 20 hr for the uncorrected data. With corrections, this minimal value was largely maintained until the end of the experiment. A lower FUV can be obtained by using maximum slopes rather than individual timepoints. The other, non-standard conditions tested showed similar qualitative dynamics, but with noise levels and timings varying between conditions as expected (*Figure 2—figure supplements 1–3*).

Although correcting for position and batch effects is essential for high-throughput experiments conducted on agar plates, there is a danger that any normalisation strategy could also create false positives. Specifically, a grid colony positioned next to a rapidly growing colony will be smaller (due to nutrient competition), leading to underestimation of the expected fitness in that area which will further increase the fitness estimate of neighbouring colonies. This argument applies equally the other way around; grid colonies positioned next to slow growers have access to more nutrients. Indeed, after reference grid normalisation, we often observed a (generally weak but detectable) secondary edge effect for colonies positioned in the next inward row/column (*Figure 1—figure supplement 2B*). We found, however, that this effect can be remedied by an additional row/column median normalisation, if the majority of strains in each row/column has no growth effect (as is usually the case when working with knock-out collections). Being a toolbox (not a black box), *pyphe* requires the user to think about their strains, choice of control strains as well as plate layout and to choose a suitable normalisation. Users have the option to perform only one of the two implemented normalisations or both (in which case grid normalisation will be done before row/column median normalisation), which allows users to tailor data analysis to their experiments.

To gauge if phenotype exaggeration globally presents a problem in other parts of the plate, we compared raw and final corrected colony sizes and maximum slopes to the respective sum of all its 8 neighbours. For uncorrected fitness values, there was generally a positive correlation (stronger for colony sizes than for slopes), indicating that regional plate effects dominate over competition between neighboring colonies. This bias was removed after correction. Importantly no negative correlation was observed. We conclude that grid correction does not lead to any significant effect exaggeration.

## Monitoring cell viability with phloxine B provides an independent and complementary phenotypic readout to growth assays

The addition of phloxine B to agar medium stains colonies in different shades of red, reflecting the fraction of dead cells, which can provide an additional phenotype readout from the same colony used for growth measurements. To investigate how colony size and redness relate, we used the *pyphe* pipeline to characterise 238 *S. pombe* single-gene deletion strains in 70 conditions in biological triplicates (n = 59,350 total colonies profiled, including controls but excluding grid colonies, *Supplementary file 2*). The two fitness proxies showed little correlation (Pearson r = −0.088) after correction of colony sizes using the grid approach with subsequent row/column normalisation and correction of redness scores by row/column median normalisation only (*Figure 3A*). Normalisation strategies for redness images are described in *Figure 3—figure supplement 1*. Many mutant-condition pairs showed a strong phenotype in only one of the two read-outs. Noise levels of redness scores were very low (CV = 1.04%) and the biological signal strong (FUV = 7.83%). We conclude that the phloxine B redness scores provide robust, precise information on mutant fitness, and serves as a

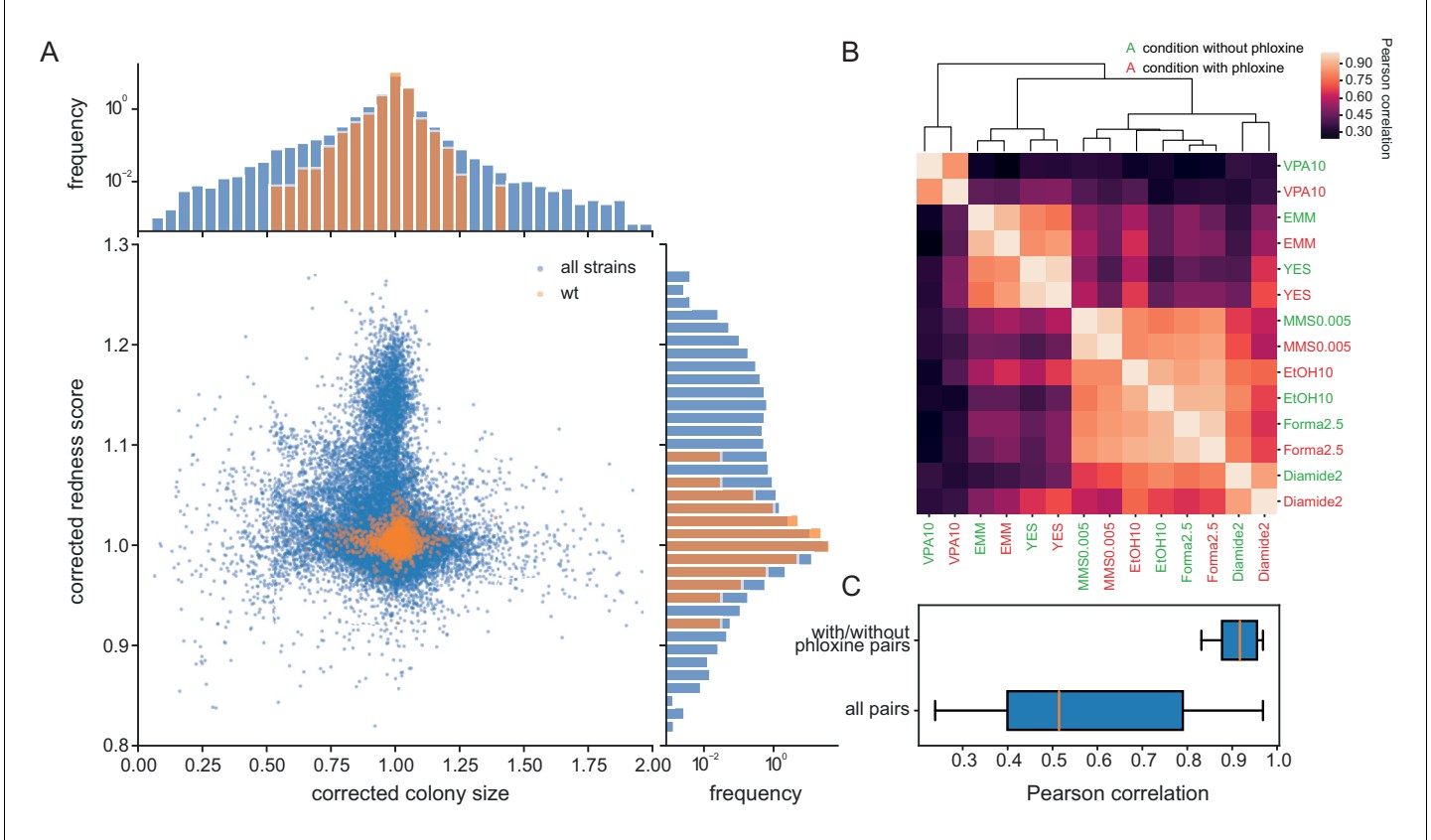

**Figure 3.** Phloxine B provides an orthogonal and independent fitness proxy. (**A**) Relative colony sizes and redness scores after correction for 238 single gene knock-outs in 70 conditions (after quality filtering as described in Methods, three biological replicate colonies for each condition-gene pair are shown individually). The two read-outs are only weakly anti-correlated (r = −0.088) and many mutant-condition pairs show a strong phenotype in only one of the two fitness proxies. Axes were cut to exclude extreme outliers for visualisation. The redness score was robust with a CV of 1.04% and a FUV of 7.83% (histogram on right). For comparison, the CV and FUV of the colony size read-out were 6.1% and 31.5%, respectively (top histogram). (**B**) Clustered Pearson correlation matrix of averaged corrected colony sizes (n = 3) for 7 conditions with and without phloxine B. Repeats with and without dye consistently cluster together indicating general robustness of our measurements across batches and no substantial mutant-condition-dye interactions. (**C**) Boxplot comparing the pairwise correlation between conditions with and without phloxine B (median = 0.92) and all possible pairs from (**B**) (median = 0.51).

The online version of this article includes the following figure supplement(s) for figure 3:

**Figure supplement 1.** Normalisation of redness data for 238 knock-out mutants.

largely orthogonal and independent measure compared to the (well correlated) growth rate or colony size measurements.

Phloxine B can be toxic if exposed to light (*Qi et al., 2011*), so we tested whether phloxine B changes growth parameters by determining colony sizes for our mutant set in 7 conditions. Measurements with and without phloxine B were performed in different batches and in different weeks to exclude that batch effects increase the correlation. Within the 14 phenotype vectors measures in total, identical conditions with and without phloxine clearly and consistently clustered together (*Figure 3B*). The median correlation for the 7 condition pairs with and without phloxine was 0.92, which was substantially higher than that of all possible pairs from the 14 phenotypes (*Figure 3C*). We conclude that the main driver of the biological signal is the condition and not whether phloxine B is included. We tested for specific gene-condition pairs showing differential growth on media with and without phloxine (*Supplementary file 3*). This analysis identified a single gene, the trehalose-6-phosphate phosphatase *tpp1*, as having a small slow-growth phenotype on rich media (ratio of medians of corrected colony sizes 0.89, $p_{adj}$ = 0.028) and a moderate effect on minimal media (ratio of medians = 0.79, $p_{adj}$ = 0.049). In order to account for such genotype-specific effects, differential fitness should generally be assessed against a control condition also containing phloxine B.

## Phloxine B staining informs about fraction of live cells in colony

Finally, we tested whether and how the colony redness score relates to the viability of cells in the colony. We determined colony composition and viability status at the single cell level using Image-Stream flow cytometry. Across 23 samples, obtained from colonies with varying redness scores (*Figure 4A*), phloxine B staining classified cells into three populations (*Figure 4B and C*, *Figure 4— figure supplement 1*, *Supplementary file 4*): live cells which showed a background level of staining, dead cells which were brightly stained, and lysed or damaged cells which showed no staining. The fraction of live cells (alive/(dead+lysed+alive)) was inversely correlated (Pearson r = −0.88, with some grouping of strains) with colony redness scores obtained with *pyphe-quantify* and row/column median corrected by *pyphe-analyse* (*Figure 4D*). This correlation was stronger than the correlation of colony redness scores with the fraction of live and lysed cells (lysed+alive /(dead+lysed+alive), r = −0.78, *Figure 4—figure supplement 2*), suggesting that lysed cells, while not stained in the FACS, do contribute to colony redness. This is explained by the dye not being washed out in

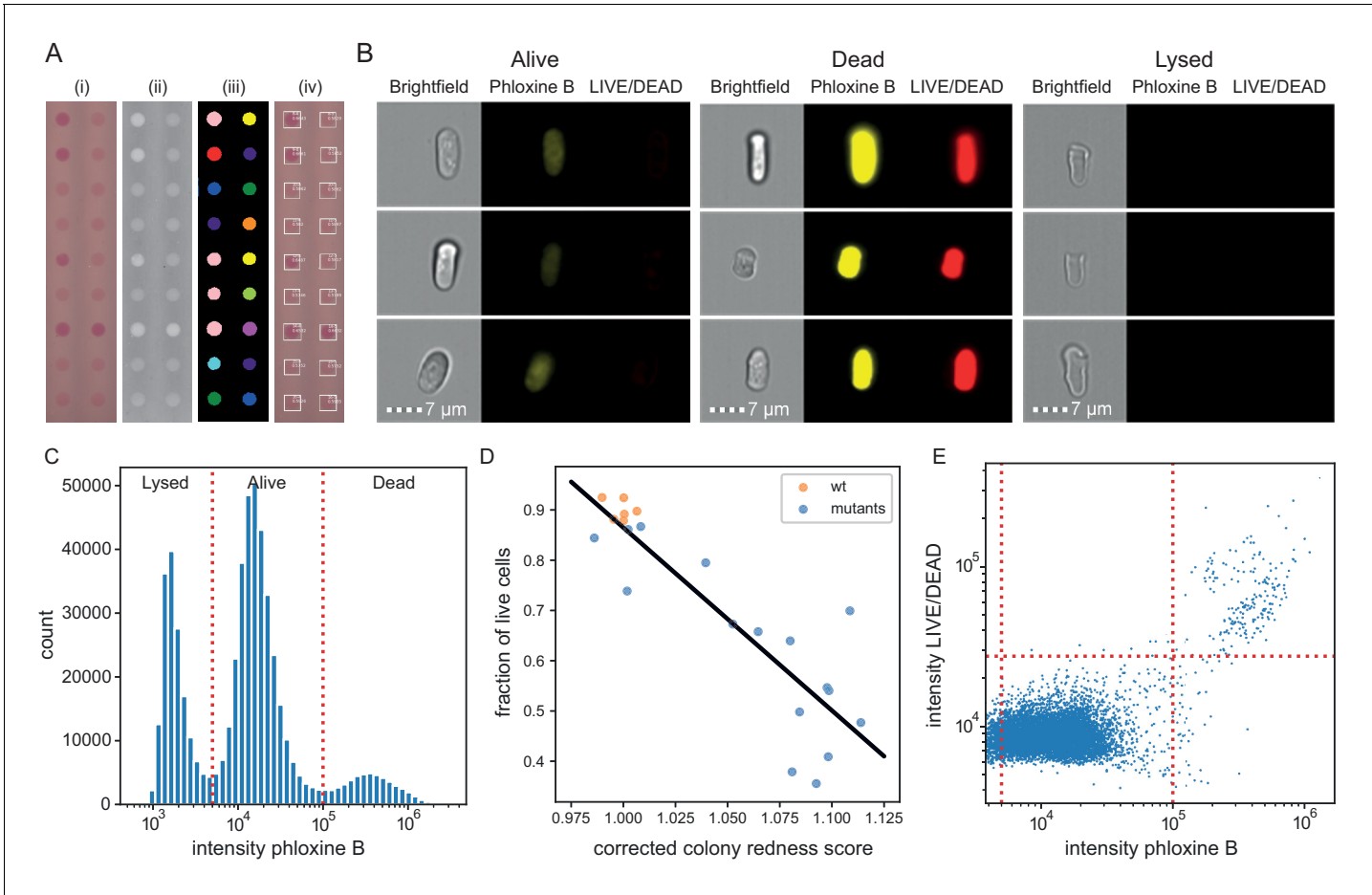

**Figure 4.** Phloxine B staining reflects percentage of dead cells. (**A**) Example of colony redness score extraction by *pyphe-quantify* in *redness* mode. From the acquired input image (i), colors are enhanced and the background subtracted (ii), colonies are identified by local thresholding (iii), and redness is quantified and annotated in the original image (iv). (**B**) Representative cells for alive, dead and lysed cells using imaging flow cytometry (ImageStream). Lysed cells show no signal in either the phloxine B or LIVE/DEAD channels. Live cells show an intermediate signal intensity in the phloxine B channel but no LIVE/DEAD signal. Dead cells are brightly stained in both channels. (**C**) Histogram of intensities in phloxine B channel across 23 samples with three populations (lysed, alive and dead) clearly resolved. (**D**) Fraction of live cells (live/(lysed+dead)) by ImageStream correlate with colony redness scores (corrected by row/median column normalisation) obtained with *pyphe*. (**E**) Co-localisation of phloxine B stain with LIVE/DEAD stain for the standard lab strain *972*. Both readouts agree with 99.3% accuracy using the illustrated thresholds.

The online version of this article includes the following figure supplement(s) for figure 4:

**Figure supplement 1.** Distribution of phloxine B intensities in ImageStream.

**Figure supplement 2.** Fraction of strongly stained cells depending on colony redness score.

colonies, unlike in cells resuspended in PBS for flow cytometry analysis. We next asked how well phloxine B staining agrees with a distinct, established dead-cell stain (LIVE/DEAD). In wild-type cells, staining with both dyes agreed closely (accuracy 99.3% using LIVE/DEAD classification as ground truth, *Figure 4E*). We conclude that phloxine B staining, combined with our imaging and analysis pipeline, provides a sensitive and accurate readout reflecting the proportion of live/dead cells in a colony.

## Redness readouts should be obtained in stationary phase

We have shown that for colony sizes similar results are obtained even if the plates are incubated for a few days after rapid growth has ended. The same is not necessarily expected for colony redness scores. In fact, colonies might appear red due to strains producing dead cells during growth or due to death when non-dividing cells reach the end of their chronological lifespan, which is temporally decoupled from growth. Certainly, if colonies are left for a very long time, cells will age, with striking physiological adaptations and eventually cell death (*Váchová and Palková, 2018*). To investigate how much the choice of timepoint matters with colony redness scores, we acquired colour images every 20 min for 48 hr on standard rich media for the set of 238 *S. pombe* single gene knock-outs. Each image from the experiment was analysed with *pyphe-quantify* in redness mode. In general, we do not recommend analysing images of young, small colonies for redness. All colonies showed a background signal unspecific to the dye and this increased with colony thickness. During early time-points, we therefore detect an increase in raw, uncorrected redness (*Figure 5A*).

Timepoints generally correlated well after rapid growth had ended for a period of at least 24 hr (*Figure 5B*). The CVs and FUVs were stable over this time as well (*Figure 5C*). These robust characteristics thus allow sufficient time for scanning without the need to hit a certain 'sweet spot'. For our work with knock-out libraries, we imaged plates soon after growth had slowed down. We identified a group of mutants with the strongest redness phenotype in the set (corrected colony redness >1.05). These colonies showed a clear and strong increase in redness during growth (*Figure 5D*), suggesting that here redness was not temporally decoupled from growth. As for colony growth, we conclude that the exact timepoint to determine colony redness is not that critical, as long as colonies are not growing rapidly anymore.

## Discussion

High-throughput colony-based screening is a powerful tool for microbiological discovery and functional genomics. Using a set of diverse wild yeast strains, we show that the fitness correction approach implemented in *pyphe* effectively reduces noise in the data. Importantly, for endpoint measurements the corrected fitness is independent of the exact timepoint, as long as a late timepoint is chosen, and late colony sizes are tightly correlated with maximum slopes of colony areas. This finding has two important implications. First, our results show that growth-rate measurements do not necessarily boost phenotyping experiments in the sense that they contain novel information, while one can compensate for the reduced precision of end-point measures by measuring more independent replicates. Second, little if anything is gained from precisely pre-defining incubation times of assay plates prior to scanning. Instead, plates can be simply incubated for longer (usually 2–5 days for fission yeast), especially if the assay condition slows down growth. By using genetically and phenotypically diverse wild strains for these experiments, we covered strains with diverse morphology and growth behaviour. However, we cannot exclude that this tight correlation does not hold true for other species of microbes.

Furthermore, we show that colony viability measured by phloxine B staining and image quantification by *pyphe* provides a largely orthogonal and independent readout to colony sizes, thus offering an additional trait for mutant profiling. Redness scores obtained with the *pyphe* pipeline closely reflect the number of live cells in the colony. We report that corrected (relative) redness scores are globally uncorrelated to corrected colony sizes in endpoint measurements of *S. pombe* knock-out mutants. The simplest explanation for how colonies can show normal growth even though a substantial fraction of its cells is dead is that growth and death are temporally uncoupled. While this does not seem to be the case for the knock-out mutants investigated, it might be the case in other scenarios, e.g. when working with wild strains. Similarly, they could be spatially decoupled. As not all cells in the colony are actively dividing, especially during later growth (*Meunier and Choder, 1999*), and

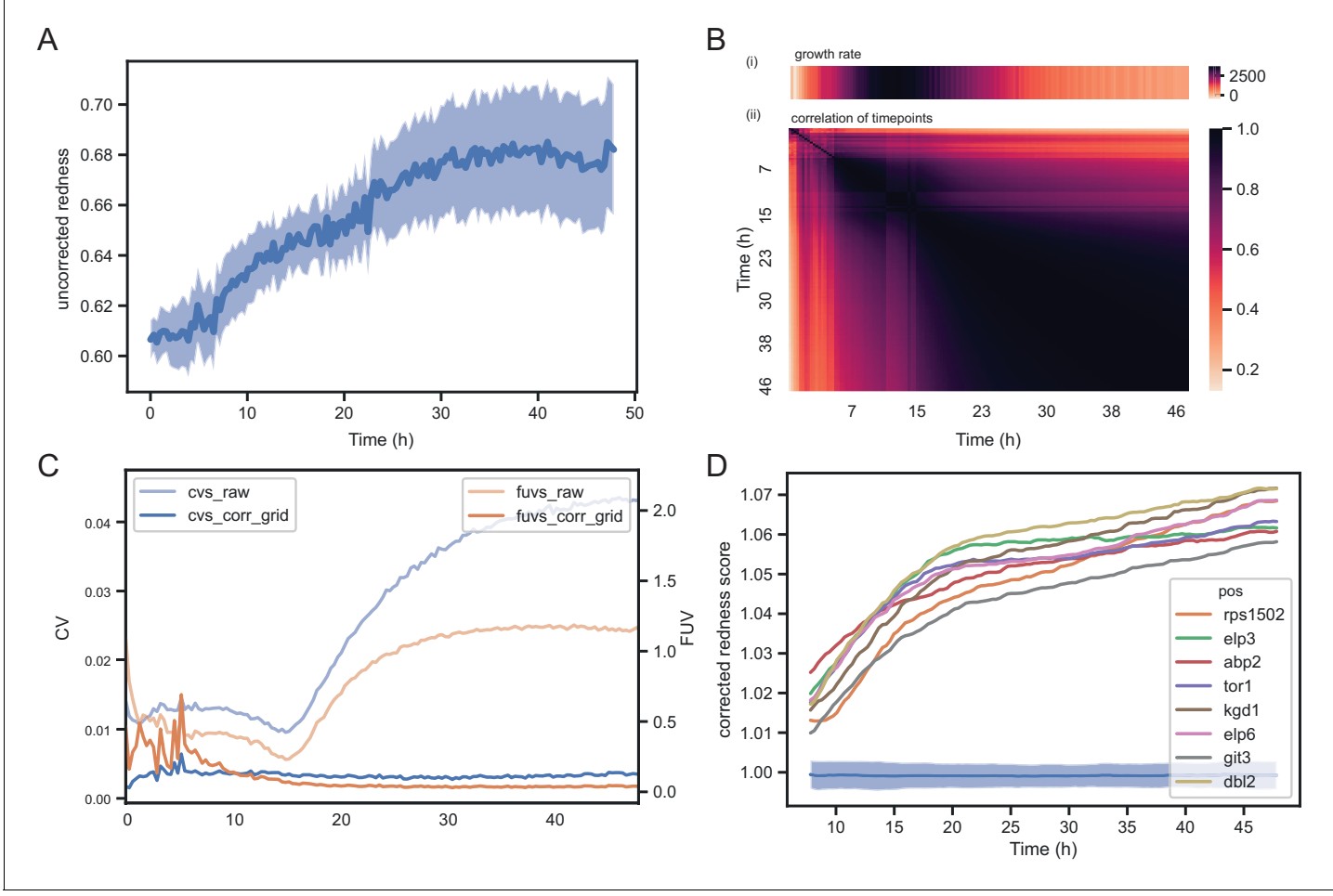

**Figure 5.** Temporal dynamics of phloxine B colony redness scores. (**A**) Raw redness scores over time for 96 wild-type grid colonies (dark line shows mean, shaded area shows standard deviation). The uncorrected redness increases as colonies grow as there is a background signal unrelated to cell death. (**B**) Correlation matrix corrected redness scores for all 238 strains over 48 hr (3 timepoints per hour). The readout is stable from the point at which fast growth ends and remains tightly correlated for at least 24 hr. (**C**) CVs and FUVs during 48 hr. Grid normalisation effectively neutralises non-biological effects. (**D**) Redness curves for selected mutants showing the strongest red phenotype. Increased redness is visible from the start, and this further increases as colonies grow. Therefore, in this case, growth and death are not temporally decoupled.

potentially in stress conditions, a subset of cells could die without the overall colony growth being affected. This idea is supported by the observed uneven distribution of redness within the colony (which we currently do not capture with pyphe). Furthermore, colonies could sustain normal growth if viability were sacrificed for growth rate (*Nakaoka and Wakamoto, 2017*). Explaining the observed disparity between redness and size data should be a priority for future research and the explanation may depend on the strains, conditions, incubation times, or technical factors (or combinations thereof). Colony redness analysis opens up new avenues of investigations, for example for high-throughput chronological lifespan experiments. It will be important to examine the relationship between redness scores and live cells if the proportion of live cells drops to very low levels as the redness signal may saturate. Potentially even more information may be contained in the distribution of dead cells within a colony, which is hard to describe quantitatively and not reported by *pyphe-quantify*.

The *pyphe* toolbox and underlying python package provide a versatile pipeline for analysing fit-ness-screen data. *Pyphe* is an all-in-one solution enabling image acquisition, quantification, batch and plate bias correction, data reporting and hit calling. *Pyphe* is flexible and accepts growth curves and endpoint measurements as well as colony sizes and staining as input. *Pyphe* functionality is provided in the form of multiple separate, simple and well-documented command line tools operating

on human-readable files. *Pyphe* is written for the analysis of extremely large data sets (thousands of plates, millions of colonies), and its modular design allows the easy integration of other, future tools and scripting/automatisation of analysis pipelines which aids reproducibility.

# Materials and methods

## Key resources table

| Reagent type (species) or resource | Designation | Source or reference | Identifiers | Additional information |
|---|---|---|---|---|
| Strain, strain background (*Schizosaccharomyces pombe*) | 57 *S. pombe* wild strains | *Jeffares et al., 2015* | JBxxx | These strains were identified as a set of most diverse strains from the overall collection |
| Strain, strain background (*Schizosaccharomyces pombe*) | 238 *S. pombe* knock-out strains | Bioneer and (*Sideri et al., 2015*) | Pombase gene IDs and names | The original library obtained from Bioneer was made prototrophic by crossing with suitable strain. Genes were selected to cover GO functional categories and include unknowns. |
| Chemical compound, drug | Phloxine B | Sigma | Cat# P2759 | Prepared as a 5 g/L (1000x) stock in water and stored at 4°C in the dark. |
| Software, algorithm | *Pyphe* | This publication | *Pyphe* provides the following tools: *pyphe-scan, pyphe-scan-timecourse, pyphe-quantify, pyphe-analyse, pyphe-interpret, pyphe-growthcurves* | Version 0.95 was used for preparation of this manuscript. |
| Other | Scanner | Epson | V800 Photo | |

## Software availability statement

*Pyphe* is open software published under a permissive license. We welcome bug reports, feature requests and code contributions through https://github.com/Bahler-Lab/pyphe. *Pyphe* is also available through the Python Package Index at https://pypi.org/project/pyphe/.

## Wild strain test data set

An overnight liquid culture of strain *972 h-* in YES medium was pinned in 96-colony (8 × 12) format on YES agar medium, using a RoToR HDA pinning robot (Singer Instruments) and grown for two days at 32°C. This grid was combined with randomly arranged plates of the 57 wild strains in 1536 (32 × 48) format and grown for 2 days at 32°C. Strains were then copied onto fresh assay plates, using the 1536 short pinning tool at low pressure. Plates were placed in scanners (EPSON V800) in an incubator at 32°C and images were acquired every 20 min for 48 hr using *pyphe-scan-timecourse*. Growth curves were extracted using *pyphe-quantify* in timecourse mode with the following settings: –s 0.1. Growth curve parameters were extracted with *pyphe-growthcurves* with the –fitrange 12 option. Individual images were analysed with *gitter* using the following settings: –inverse TRUE –remove.noise TRUE. Grid correction of maximum slopes and individual timepoints was performed in *pyphe-analyse*.

## Knock-out test data set

238 mutants, broadly spanning GO Biological Function categories plus several uncharacterised genes, were selected from a prototroph derivative of the Bioneer deletion library (*Sideri et al., 2015*). Strains were arranged in 384-colony (16 × 24) format with a single 96 grid placed in the top

left position, so that the grid includes one colony in every fourth position within the 384-colony array. To prepare replicates, this plate was independently pinned 3 times from the cryostock on solid YES media for each batch. From these plates, colonies were then spotted on assay plates containing various toxins, drugs or nutrients. The conditions used in *Figure 3B* are: EtOH10 is YES+10% (v/v) ethanol, VPA10 is YES+10 mM valproic acid, MMS0.005 is YES+0.005% (v/v) methyl methanesulfonate, Forma2.5 is YES+2.5% (v/v) formamide, Diamide2 is YES+2 mM diamide, EMM is standard Edinburgh Minimal Medium, YES is standard Yeast extract with supplements and 3% glucose. Assay plates were usually grown for 2 days at 32°C but this varied according to the strength of the stress slowing the growth of the colonies. After incubation, images were acquired using EPSON V800 scanners and *pyphe-scan* and quantified with *gitter* (see options above) or *pyphe-quantify* in *redness* mode. Grid correction and subsequent row/column median normalisation of maximum slopes and individual timepoints was performed in *pyphe-analyse*. Row/column median normalisation was applied to redness data plates. For the size data set, 0-sized colonies and colonies with a circularity below 0.85 were set to NA. Plates with a CV > 0.2 or FUV >1 were removed as those most likely represent conditions in which the stress was too strong or where technical errors occurred.

## Imaging flow cytometry

We picked 23 colonies with varying redness from the collection of 238 *S. pombe* deletion strains grown on solid YES with 5 mg/L phloxine B for 3 days at 32°C and resuspended in 1 mL of water. For analysis of phloxine B staining, 500 µL of this cell suspension were centrifuged at 4000 g for 2 min, the supernatant was removed and the pellet resuspended in 75 µL of PBS. For analysis of phloxine B and LIVE/DEAD co-staining, 500 µL of the same suspension were centrifuged at 4000 g for 2 min, the supernatant was removed and the pellet resuspended in 300 µL of LIVE/DEAD solution (LIVE/DEAD Fixable Far Red Dead Cell Stain Kit, for 633 or 635 nm excitation, ThermoFisher Scientific, Cat. no. L34974). LIVE/DEAD solution was prepared according to manufacturer's instructions (1:1000 dilution in H2O from a stock solution dissolved in 50 uL of DMSO). The pellet was resuspended and incubated for 30 min in the dark. Cells were then spun down and resuspended in 75 µL of PBS.

Immediately prior to analysis, samples were sonicated for 20 s at 50W (JSP Ultrasonic Cleaner model US21), and transferred to a two-camera ImageStreamX Mk II (ISX MKII) imaging flow cytometer (LUMINEX Corporation, Austin, Texas) for automated sample acquisition and captured using the ISX INSPIRE data acquisition software. Images of 5000–12,000 single focused cells were acquired at 60x magnification and low flow rates, using the 488 nm excitation laser at 90 mW to capture phloxine B on channel 3; 642 nm excitation laser at 150 mW to capture LIVE/DEAD cells on channel 11; bright field (BF) images were captured on channels 1 and 9, and side scatter (SSC) on channel 6. For co-stained cell analysis, to generate a compensation matrix, cells stained either with phloxine B or with LIVE/DEAD dye individually were captured without brightfield illumination (BF and SSC channels were OFF). The compensation coefficients were calculated automatically using the compensation wizard in the Image Data Exploration and Analysis Software (IDEAS) package (v6.2). Populations of interest (single focused cells) were gated in IDEAS and the features of interest (dye intensities) were then exported for further analysis using Python. Intensity values were subtracted by their minimum over all samples (which was slightly below zero) and added to 1 prior to log10 transformation. Thresholds for the three populations were set manually based on the intensity histogram across all samples.

## Redness timecourse dataset

The mutant collection was woken up from the cryostock on YES media and copied onto fresh YES with 5 mg/L phloxine B. Images were acquired every 20 mins with *pyphe-scan-timecourse*. Images were analysed with *pyphe-quantify* redness with –s 0.1. Timepoints were grid corrected using *pyphe-analyse*.

## Acknowledgements

Mimoza Hoti helped with the wet lab part of phenotyping the 238 knock-out mutants. This research was funded by Wellcome Trust Senior Investigator Awards to JB [grant number 095598/Z/11/Z] and to MR [grant number 200829/Z/16/Z], as well as a BBSRC Project Grant to JB [grant number BB/

R009597/1]. This work was also supported by the Francis Crick Institute which receives its core funding from Cancer Research UK (FC001134), the UK Medical Research Council (FC001134) and the Wellcome Trust (FC001134).

## Additional information

### Funding

| Funder | Grant reference number | Author |
|---|---|---|
| Wellcome | 095598/Z/11/Z | Stephan Kamrad<br>María Rodríguez-López<br>Cristina Cotobal<br>Jürg Bähler |
| Wellcome | 200829/Z/16/Z | Stephan Kamrad<br>Clara Correia-Melo<br>Markus Ralser |
| Biotechnology and Biological Sciences Research Council | BB/R009597/1 | María Rodríguez-López<br>Jürg Bähler |
| Medical Research Council | Francis Crick Institute FC001134 | Stephan Kamrad<br>Clara Correia-Melo<br>Markus Ralser |
| Wellcome Trust | Francis Crick Institute FC001134 | Stephan Kamrad<br>Clara Correia-Melo<br>Markus Ralser |
| Cancer Research UK | Francis Crick Institute FC001134 | Stephan Kamrad<br>Clara Correia-Melo<br>Markus Ralser |

The funders had no role in study design, data collection and interpretation, or the decision to submit the work for publication.

### Author contributions

Stephan Kamrad, Conceptualization, Resources, Software, Formal analysis, Investigation, Visualization, Methodology, Writing - original draft; María Rodríguez-López, Conceptualization, Resources, Validation, Investigation, Methodology, Writing - review and editing; Cristina Cotobal, Resources, Validation, Investigation, Methodology, Writing - review and editing; Clara Correia-Melo, Investigation, Methodology; Markus Ralser, Conceptualization, Supervision, Funding acquisition, Project administration, Writing - review and editing; Jürg Bähler, Conceptualization, Supervision, Funding acquisition, Methodology, Project administration, Writing - review and editing

### Author ORCIDs

Stephan Kamrad (iD) https://orcid.org/0000-0002-5957-4661
María Rodríguez-López (iD) https://orcid.org/0000-0002-2066-0589
Cristina Cotobal (iD) https://orcid.org/0000-0002-5877-2228
Clara Correia-Melo (iD) https://orcid.org/0000-0001-6062-1472
Markus Ralser (iD) https://orcid.org/0000-0001-9535-7413
Jürg Bähler (iD) https://orcid.org/0000-0003-4036-1532

### Decision letter and Author response

Decision letter https://doi.org/10.7554/eLife.55160.sa1
Author response https://doi.org/10.7554/eLife.55160.sa2

## Additional files

### Supplementary files

• Supplementary file 1. Corrected maximum slopes and endpoints for 57 wild strains in 8 conditions.

- Supplementary file 2. Relative redness scores and colony sizes for 238 knock-out mutants.
- Supplementary file 3. Differential fitness of 238 knock-out mutants in conditions with and without phloxine B.
- Supplementary file 4. ImageStream classification counts for mutants.
- Transparent reporting form

## Data availability

Relevant datasets are included as Supplementary Files 1-4. Please see the *pyphe* github repository for example data illustrating the use of *pyphe* (https://github.com/Bahler-Lab/pyphe).

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

## Appendix 1

# Image quantification algorithms

*Pyphe-quantify* is a command line tool for the analysis of images containing microbial colonies based on *scikit-image* (**van der Walt et al., 2014**). By default, it analyses all. jpg image files in the directory where it is executed. Alternatively, the user can set a pattern to specify input images and all image formats supported by *scikit-image* can be used (e.g. tiff, jpg, png). *Pyphe-quantify* can operate in three distinct modes: batch (analyse colony areas in each image separately), redness (analyse colony redness in each image separately) and timecourse (analyse colony area and thickness in a stack of images from a timeseries) mode. *Pyphe-quantify* produces simple csv files (one per plate for batch and redness mode) which can be directly processed further by *pyphe-analyse* or analysed using other software.

Common to all three modes is the need to match identified colonies in the image to their row-column position, which is configured using the –grid option. *Pyphe-quantify* implements an automatic grid detection (used by setting –*grid auto_384* or –*grid auto_1536*) which identifies peaks in rows and columns pixel intensities. This is done using the find_peaks function from scipy's signal module (using the image dimensions and number of colonies per row/column to define a suitable minimal distance between peaks). The distance between peaks is then determined using an outlier-robust, trimmed mean. The maximum pixel position of the first colony is then determined as (image dimension - (mean distance between colonies * (colonies per row/column - 1)). A cosine function with the same periodicity as the distance between colonies is created. The fit of that function to the data (row/column mean intensities) is then evaluated by taking the sum of squared differences for each possible pixel offset from 0 to the maximum previously determined. The expected positions of all colonies are then computed using the start position and the mean distance.

Automatic grid detection is in our hands the most common reason for image analysis tools to fail. We have therefore given the user full control over defining expected colony position. If all plates were scanned with the same fixture taped firmly to the scanners, colony positions are actually highly consistent across images. This means grid definitions can be 'hard-wired'. The fixture provided by us has been preconfigured and is done using the -*grid pp_384* or –*grid pp_1536* options. Simultaneously, with the goal of maximum flexibility, *pyphe-quantify* offers the possibility of manually defining grid positions. In that case, the argument has to be in the form of 6 integer numbers separated by '-':<number of rows>-<number of columns>-<x position of the top left colony>-<y position of the top left colony>-<x position of the bottom right colony>-<y position of the bottom right colony>. Positions must be integers and are the distance in number of pixels from the image origin in each dimension (x is width dimension, y is height dimension). The image's origin is, in line with *scikit-image* convention, in the top left corner. Getting those coordinates is trivial and can be done, for example in Microsoft Paint. The option to manually define grid positions is important in our experience, as automatic gridding is the step where most image analysis tools typically fail (especially if plates have many missing colonies or images are rotated).

In *batch* mode, *pyphe-quantify* will analyse colony sizes in each image individually. First, morphological components are identified by thresholding the image. By default the Otsu method is used to find the threshold (**Otsu, 1979**), but this can be tuned by the user by providing a coefficient to be used with this threshold or an absolute threshold. Components are then filtered by size to exclude erroneous identification of small particles, such as dust, as colonies. Border components are removed. Components are then matched to grid positions. By default, a component is assigned to a particular position if it is less than a third of the distance between two grid positions away from a position. This threshold can be set by the user. This means that in case of missing colonies, there will be no data for the corresponding grid correction, and this position will be missing from the output file (i.e. it will not be 0). Similarly, two components can be assigned to the same grid position in the case of contaminations. This can be disabled by the user to retain only the component nearest to the grid position. When reporting all colonies, *pyphe-quantify* can be used for plates with non-

arrayed colonies. An output table is saved which contains colony area, mean intensity (an estimator that reflects thickness), circularity, perimeter and centroid coordinates. Area measurements are in very close agreement with those obtained with *gitter* (***Figure 1—figure supplement 1A***). Mean intensity measurements reported by *pyphe-quantify* are dependent on colony area as colonies get thicker as they grow (***Figure 1—figure supplement 1B***). *Pyphe-quantify* exports a qc image for every image analysed indicating the identified colonies and their assigned grid positions.

In *timecourse* mode, the final image of the time course is analysed as described above and the obtained mask (indicating the position of each colony) is then applied to all previous images of the timecourse. The background subtracted sum of pixel intensities (the mean intensity times the number of pixels) for all images combined, that is the growth curves, is reported in a single file.

Finally, *pyphe-quantify* can analyse colony redness (***Figure 1—figure supplement 1C***). Phloxine B stains dead cells within the colonies and these are usually not homogeneously distributed upon close inspection. However, for simplicity, we have developed an image analysis workflow which extracts the mean redness of colonies in high-density arrays, providing a single quantitative readout. We decided to use reflective scanning with our Epson V800 scanners (implemented in *pyphe-scan*). This is fast and produces images with consistent properties, but with the caveat that the focus position is just above the scanner glass and colonies are therefore somewhat out of focus. Additionally, there is a strong, uneven background signal from the media and colour artefacts (appearing as bright stripes between colony columns) which required a different image analysis approach. The images are first adjusted to make colony redness more visible by multiplying the red, green and blue channels by 0, 0.5 and 1, respectively and their sum is taken to produce single-channel/grayscale images. The background value for each pixel is estimated by blurring the image with a Gaussian filter with a standard deviation of the number of pixels in the image divided by 10000. The background is subtracted from the image which is then inverted. Colonies are detected by local thresholding and processed further as described for *batch* mode above. The mean intensity for each colony is computed from the processed image and reported in a similar file as described for *batch* mode. The produced QC images allow to verify grid placement and visualise the colour readout on the actual image.

**Appendix 2**

## Spatial correction algorithms

This text describes the steps performed by *pyphe-analyse* during the analysis of a typical batch of plates. All functions and objects are also available for use as a python package. At the core of *pyphe* is the Experiment object which is initialised from the Experimental Design Table (EDT). The EDT is first checked for obvious errors, including the uniqueness of plate IDs and paths to data files and if these files exist. Data from the image analysis output files is then loaded using appropriate parsers. Layout files are loaded if set by the user.

Spatial normalisation is then performed if requested by the user. *Pyphe* implements a grid correction procedure similar to that used previously (*Zackrisson et al., 2016*). In that paper, the authors use 1536 format arrays and place a 384 grid in the top left position of the plate. This means one quarter of plate positions are taken up by the grid. It also creates a problem because the right and lower edge of the plate are not covered by the reference grid. We have developed a small improvement of this technique by placing two 96 grids in opposite (top left and bottom right corners, *Figure 1—figure supplement 2A*). This leaves only two small corners of the plate (bottom left and top right) not covered by the interpolated grid surface. We solve this by extrapolating the grid by estimating the theoretical colony size of a grid strain in those corners using a linear model and the colony sizes of the two neighbouring grid colonies as input. Model parameters are determined for each experiment based on all plates using regression. We typically achieve accuracies of >90% (*Figure 1—figure supplement 2C*). This allows us to use grid correction over the entire plate without loss of data. *Pyphe-analyse* specifically looks for grid colonies with a colony size zero (this is reported by *gitter* if no colony has been detected), flags those as pinning errors and marks all neighbouring colonies NA in the final output. The grid correction itself is done using scipy.interpolate's griddata function, fitting a piecewise cubic, continuously differentiable, approximately curvature-minimizing polynomial surface to the grid positions (real grid positions and extrapolated corners). The surface created in that way represents the expected fitness for each position if the strain growing there was the same as the grid strain. The observed fitness is then divided by this expected value, producing the corrected and relative (to the grid strain) fitness of each colony.

We have noticed that doing the grid correction this way slightly over-corrects for the edge effect for strains in the row neighbouring the edge (*Figure 1—figure supplement 2B*). This is because the edge effect is usually restricted to the outermost edge only. But the values of the reference grid in the next row/column will be most strongly determined by the grid colony on the edge, leading to an over-correction (underestimation of fitness) in these positions. We therefore often perform an additional row/column median normalisation after grid correction to remedy this. For this, *pyphe-analyse* computes median values for each row and column and divides the data by both. The data is then re-scaled to a median of 1 by dividing by the overall plate median. Note that a median correction is not valid if the median is not a good estimate of the null effect. For this reason, we strongly discourage row/column median normalisation for plates in 96-colony format (where the median is computed from only 8 or 12 values). If plates contain a large number of slow or fast growers a median normalisation is also unsuitable, especially if these are distributed non-randomly in the plate. For work with knock-out libraries, where most gene knock-outs have no effect in any given condition, and with wild strains which show a median-centred distribution of subtle growth phenotypes, the additional row/column median normalisation effectively neutralises the secondary edge effect.

In some cases, the normalisation procedures can lead to artefacts. Grid normalisation can result in negative corrected fitness values if grid colonies are very small in a region (*Figure 1—figure supplement 2D*). Row/column median normalisation can produce infinite values if more than half the colonies in a row/column have size 0. These artefacts are detected by *pyphe* and set to NA.

*Pyphe* gives the option to produce QC plots for each plate in the experiment in which case a pdf file will be generated containing heat maps for all numerical data associated with that plate. Finally, all data is collated in a single table which contains position information, layout

information, all metadata provided by the user in the EDT, raw and corrected fitness values, and details about the grid correction.

**Appendix 3**

## Hit calling with *pyphe-interpret*

*Pyphe-interpret* is a tool for statistical analysis of fitness data. It takes data reports produced by *pyphe-analyse* (or other data in a suitable tidy format) which contain a single line per colony, listing strain, condition and fitness information. The column names in which to find each of these are set by the user. The tool first checks the input and prints a summary to the command line listing the number of strains, conditions, plates and total number of data points. Next, QC filters based on circularity and 0 fitness are applied if set by the user. The number of excluded data points is reported. The tool then produces a table which lists all replicates for each strain-condition pair in wide format (see documentation folder on GitHub for an example). This table (which is also exported in csv format) allows to perform t-tests highly efficiently in a vectorised manner using the ttest_ind function for masked arrays from the scipy's mstats_basic module. *Pyphe-interpret* requires the user to define a grouping (variable to use as the grouping variable for t-test) and the 'axis' column across which to apply multiple t-tests. This may initially seem complicated but enables the use of *pyphe-interpret* in two distinct scenarios: (1) Check for each condition separately (–grouping_column <condition_column>) if there is a significant difference in means between a mutant strain and a control strain (–axis_column <strain_id_column>); or (2) check for each strain separately (–grouping_column <strain_id_column>) if there is a significant difference in the means of the strain in the assay condition *vs* the control condition (–axis_column <condition_column>). We normally use the second option as it tests for condition-specific growth differences and it does not return significant results if a strain is consistently faster or slower growing than the grid strain. We use Welch's t-test (equal_var = False) as this does not assume homogeneity of variances. P-values are corrected by the Benjamini-Hochberg method across the specified axis (i.e. across all strains in scenario 2 or across all conditions in scenario 1). The tool produces a table listing summary statistics (mean_fitness, mean_fitness_log2, median_fitness median_fitness_log2, observation_count stdev_fitness) and a statistical assessment of differential fitness (mean_effect_size, mean_effect_size_log2, median_effect_size, median_effect_size_log2, p_Welch p_Welch_BH p_Welch_BH_-log10).

**Appendix 4**

## Growth curve analysis with *pyphe-growthcurves*

*Pyphe-growthcurves* is a simple tool for non-parametric growth curve analysis. It was written to directly use data produced by *pyphe-quantify timecourse* as input but other types of growth data can normally easily be adapted. There are various packages dedicated to growth curve analysis (*Fernandez-Ricaud et al., 2016*; *Kahm et al., 2010*; *Veríssimo et al., 2013*) which have more functionalities, but the goal here was to provide a simple solution integrated into *pyphe* which works well with data typically handled within the *pyphe* pipeline. The input data needs to be in csv format and contain one column per growth curve containing the population sizes in the right order (top to bottom). The first column must contain the timepoints and those must be numerical (i.e. not '1 hr', '2 hr', but 1.0, 2.0). A single input csv is analysed every time *pyphe-growthcurves* is run. Maximum slopes of growth curves are determined by linear regressions to a sliding window of size d. The regression with the highest slope is retained and the slope is reported together with the timepoint at which it occurred (center of window), the $R^2$ of the regression, as well as its y- and x-intercepts. Lag phases can be determined by the absolute or relative method. The absolute method simply returns the timepoint at which the population size crosses the user-defined threshold. For the relative method (which is the default), the average of the first n timepoints is taken as the initial biomass. The algorithm then returns the timepoint at which the population sizes exceeds p*initial biomass, where p defaults to 2 (i.e. the time taken for the first population doubling is reported). Please note that no interpolation of population sizes between timepoints is currently implemented, but simply the timepoint where the threshold is crossed is returned. A table in csv format is created which lists all above-mentioned parameters and this can be read directly into *pyphe-analyse*. If the –plots option is set, the tool produces a pdf showing all growth curves and visualising the extracted parameters.

## Appendix 5

# Plate handling protocol

## Materials and reagents

- Sterile yeast medium with and without 2% agar (we preferentially use YES or EMM for *S. pombe*)
- Serological pipette and pipette pump
- Rectangular plates (PlusPlates, Singer Instruments)
- RoToR pin pads (96 long, 96 short, 384 or 1536 short)
- 96 well sterile plates
- Phloxine B (Merck)

## Equipment

- Laminar Flow Cabinet
- Microwave oven
- Incubator
- Pinning robot (RoToR HDA, Singer Instruments or similar)
- Scanner (Epson Perfection V800) connected to Linux computer
- Fixture to hold plates in place on scanner (cutting guide available at www.github.com/Bahler-Lab/pyphe)

# Procedure

## Overview

The grid strain is prepared to grow in 96-format plates to make grid plates. Grid plates are combined with library plates to make combined plates. Combined plates are copied onto fresh agar plates to make source plates. Assay plates (containing treatments of interest) are inoculated from source plates. Assay plates are imaged and analysed further.

## 1. Plate pouring

a. Heat media in microwave with occasional mixing until completely melted. Let the media cool to approximately 60˚C.
b. Warning: Superheated agar media can pose a serious risk. Proceed carefully, never heat sealed containers and wear appropriate protective equipment.
c. If drugs are to be added to the media mix them in the media before pouring.
d. For phloxine B assays, add this reagent at a final concentration of 5 mg/L prior to pouring. Note that phloxine B is sensitive to light so it is advisable to store and incubate plates in the dark wherever possible. Phloxine B is also sensitive to oxidising agents and therefore incompatible with such assay conditions.
e. Tip: A 1000x aqueous stock of phloxine B can be kept in the fridge in the dark for several weeks.
f. Place plates on a flat surface in the sterile hood and pipette 40 ml of media in each.
g. Tip: Take up 5 ml more than required to avoid bubble formation. If bubbles occur, remove them by sucking them back up into the pipette.
h. Let plates dry for approximately 30 min. Correct dryness is important, if the plates are wet colonies will diffuse into agar.

## 2. Plate storage and handling

a. Drugless plates made to preserve or wake up collections can be stored in the fridge for a week, but plates should be removed from the fridge and let them acclimate to room temperature prior to any experiment.
b. Plates containing phloxine B should be stored in the dark, as phloxine B oxidizes in the presence of light.
c. We recommend preparing assay plates containing drugs on the day of the experiment or the evening prior to the experiment taking place. If this is done, store them appropriately

and keep them well wrapped to prevent uneven drying, and upside down to prevent condensation forming on the surface of the media. Let the plates reach room temperature before pinning.

## 3. Preparation of source plates: where to locate your grid, controls and your test strains

a. Preparation of the grid plate

i.   From a cryostock, streak out the strain that will be used as the control 'grid strain' on agar media (can be in a conventional Petri dish, add appropriate antibiotics if required). We advise to pick a standard strain which makes the comparative fitness value obtained in the end easily interpretable, for example the background strain in case of mutant collections. In general, the grid strain's fitness should not be extreme (much higher or lower) than the strains to be assayed.

ii.   Grow until colonies suitable for picking have formed (approximately 2 days at 32℃ for *S. pombe*).

iii.   Inoculate one colony of the grid strain into 30 ml liquid media (e.g. YES in the case of the standard 972 *S. pombe* strain) and grow for ~24 hr with shaking.

iv.   Pour the grid strain culture into an empty PlusPlate and use the RoToR robot to pin onto solid agar media in 96 format using 96 long pin pads.

v.   **Tip**: Make several copies as needed. You can pin approximately 10 times from one grid source plate.

vi.   Wrap the plates in cling film and place upside-down in an incubator to avoid condensation over the colonies. If the plates are not properly wrapped they will dry unevenly on the edges and will not be suitable.

vii.   Grow up for approximately 2 days until suitable colonies for pinning have formed.

b. Preparation of the library to assay

i.   If you are starting from an established library

   1.   If you are using an established yeast library stored in 96- or 384- well format, wake up the library onto the appropriate selective agar media using the RoToR and let it grow until colonies are visible at the appropriate temperature (32℃ for the *S. pombe* Bioneer deletion collection).
   2.   Once the colonies are grown, refresh the plates onto selective media the same day that you prepare your grid strain plates and let them grow for up to 2 days at the appropriate temperature.

ii.   If you are arranging your own library

   1.   Prepare fresh colonies of the strains that will be used on solid agar media plates.
   2.   Design your library layout. Every plate should contain several wild type controls (at least 10). Plates should contain no or few empty spots, but do include a footprint to mark plate number, orientation and to serve as a negative control. Fill up the rest of the positions with your assay strains and include extra replicates to fill up the plate if required.
   3.   **Tip**: If possible, we recommend to include some positive control strains (that are known to be resistant or sensitive to the stresses to be tested) in the library.
   4.   The same day that you will be starting the liquid culture of your grid strain, fill a 96 well plate with the appropriate liquid media. Inoculate each well from a colony, according to your layout.
   5.   This 96 well plate can be incubated in a stationary incubator with the lid on for ~24 hr at the appropriate temperature.
   6.   As with the grid strain plate, use the RoToR robot to pin this plate onto solid agar media using 96 long pin pads. Use vigorous mixing (in 3D, 4 cycles) of the source plate.
   7.   Wrap the plates in cling film and incubate for 2 days upside down at the appropriate temperature.

c. Preparation of the *pyphe*-ready source plates

i.   Combine your grid plate and library plates on a solid agar medium.

   1.   If your assay plates should be 1536 format
   2.   Refer to *Figure 1—figure supplement 2A* for an illustration of the arrangement process. Using 96 short pin pads and the manual programming mode of the RoToR robot, prepare

your combined plates by copying the 96 well plate containing the grid strain in the top left and bottom right corner as well as in an additional position in the middle (we normally use the C2 position). Fill the remaining 13 positions with library plates. Record exactly which library plate was used to fill each position and use this information to prepare a layout table of your assay plates.

3. **Tip**: This program can be saved and reused.
4. If your assay plates should be 384 format
5. If you want to work in 384 format, place one grid in the top left corner of each plate. Note that you will lose grid-corrected phenotypes for colonies on the bottom and right edge because these are not covered by the grid. You will also not have a control grid to check the quality of the grid correction. It is usually preferable to use 1536 format with more repeats, even if you have few strains.

ii. Grow for 1 or 2 days, wrapped in cling film and upside down in an incubator at the appropriate temperature.

iii. Copy your combined plates onto fresh plates to make your *pyphe* source plates. This will even out any differences in growth from different inoculum amounts from the previous steps and create a more even spacing of colonies.

   1. **Tip**: You need to make several copies if you have a large number of assay plates/conditions to be tested. As a rule of thumb, you can use the same plate for pinning ~6 plates on 1536 format ~8 plates on 384.

iv. Grow for 1 or 2 days at the appropriate temperature (but keep it consistent). At this stage the plates are ready to be used in the assay.

## 4. Phenotyping with the *pyphe* pipeline

a. Using the appropriate 384 or 1536 short pin pad, inoculate your source library plates onto your assay plates using the RoToR robot. Label your plates clearly with the replicate number, plate layout and condition.

b. Tip: Use low pressure (around 10% for 384 plates and 4% for 1536 format plates) in order to get a small, consistent inoculum.

c. Tip: Check every time that you did not miss to pin an area of your plate. If this happens, repeat the pinning using a fresh, spare assay plate. If this happens repeatedly, you assay plates were not prepared on a flat, level surface or dried out unevenly.

d. Tip: We recommend using the random offset for picking up the colonies.

e. Wrap the assay plates on cling film and incubate upside down at the appropriate temperature on an incubator. For 1536 plates and mild stressors around 18 hr incubation is enough time for phenotype observation, 384 format or higher stressors might require further incubation times.

f. Proceed with image acquisition and data analysis. See manuals and help on GitHub for this. We recommend preparing the Experimental Design Table (which will be later required by *pyphe-analyse*) during scanning, making note of all relevant data and meta-data associated with each plate. The table should contain columns for condition, plate layout, image location, incubation time, batch and scan/pin dates. Save this table in CSV format.

Tip: For large screens containing several batches, consistent naming is essential. We usually define a condition shortcut in a separate table and include the dose without units for brevity, for example an entry in the condition column in the EDT may state 'VPA10' which is short for YES+10 mM valproic acid.

Tip: File paths should generally not contain any spaces, non-standard characters or characters forbidden in Unix or Windows file names. Name your condition shortcuts, layouts and replicates accordingly.

Tip: Comments or observations which may be important for later analysis (e.g. if there were pinning errors or other issues) should be included in an extra column.

Tip: Any additional (meta-)data can and should be included and will be carried through to the data report produced by *pyphe*.

