## [Decision Letter]

**Acceptance summary:**

We believe that the *Pyphe* toolbox will prove a valuable tool for the community and will help set standards for image analysis of microbial growth and physiology.

**Decision letter after peer review:**

Thank you for submitting your article "*Pyphe*: A python toolbox for assessing microbial growth and cell viability in high-throughput colony screens" for consideration by *eLife*. Your article has been reviewed by three peer reviewers, including Kevin J Verstrepen as the Reviewing Editor and Reviewer #1, and the evaluation has been overseen by Aleksandra Walczak as the Senior Editor. The following individual involved in review of your submission has agreed to reveal their identity: Jonas Warringer (Reviewer #2).

The reviewers have discussed the reviews with one another and the Reviewing Editor has drafted this decision to help you prepare a revised submission.

We appreciate your efforts to assemble a streamlined pipeline for image-based high-throughput microbial growth assays. We believe that such a pipeline would potentially be of interest to the broad readership of *eLife*. However, reviewer 2, who is an expert in this area and was actually one of the suggested reviewers, identified several technical issues that need to be resolved. We realize that we are requesting quite a bit of additional work, but all reviewers unanimously agreed that these are crucial. We realize that we are requesting quite a bit of work to further improve the pipeline, and we are not sure whether it will be possible for you to address all these comments in a timely manner. However, we decided that since the core idea of the pipeline is very valuable and would in principle be a good fit for *eLife*, we would leave the decision whether to address our concerns and resubmit a revised version with you.

Essential revisions:

1) All reviewers agree that the pipeline in itself is not novel, and it would be advisable to stress and acknowledge more how the pipeline builds upon previous work and similar pipelines

The following comments were directly taken from the report of reviewer 2, but reviewer 1 and 3 agree that these need to be addressed.

2) As an expert user, reviewer 2 has issues with core aspects of the method.

A) The v800 model now seems out of production. For the *Pyphe*-*scan* method to be useful for other labs, clearly it must work with other scanners. At the very least, the authors should show that the pipeline works and provides equivalent data with the 850 model.

B) In this context, it seems unfortunate that authors have chosen not to implement the pixel calibration function of the Zackrisson et al., 2016 approach. This uses a grey-scale calibration strip attached to each fixture to ensure that registered pixel intensities becomes comparable across types of instruments, scanners, external conditions and time – the latter is quite important as the properties of light sources and light receptors change as a function of age.

C) The authors have chosen to, as a default, use a lossy, irreversible compression format for their images, JPG. This format uses very inexact approximations and discards much data when representing the original content of the image. JPG is not recommended when quantitative data is to be extracted from an image. The use of e.g. TIFF is much to be preferred: if the authors want to maintain the JPG format, a minimum requirement is that they show that colony delineations, pixel intensity and background intensity are not affected by the lossy compression format. Otherwise, it should be discarded and no data based on it included.

D) Quality control, QC, is normally a substantial investment of time. It was not clear from the paper how the QC or the QC interface works. If an automated, or at least a semi-automated process is not implemented this is a serious shortcoming and would be another step backwards from the Zackrisson et al., 2016. The importance of a high throughput QC becomes clear when studying the example growth curves supplied together with the software: it is apparent that a large fraction of the slopes extracted are wildly inaccurate, or indeed pure noise, and does not reflect the actual growth. How is this handled? If the user has to filter these out manually, with no software guidance, the pipeline is no longer high-throughput.

E) The graphical interface provides user-friendliness. However, the GUI seems to only be applicable to the later stages of the pipeline? The initial steps are command-line based, which clearly does not square with claims of user-friendliness in general. As an example: to position the grid, the user is requested to input pixel coordinates, as integers, from the image in the command-line interface. This lack of a GUI for the early steps is another step backwards from the graphical interface of plate position or automatic grid detection implemented in Zackrisson et al., 2016?

F) How does *Pyphe* deal with the identification of scanners, when multiple scanners are connected to a computer? Explicitly: how does *Pyphe* in these conditions ensure that the images comes from the scanner that the user thinks the image comes from? In the CLI interface the scanner number can be input, but it is stated that this may change if the scanner is turned off. Such a change will supposedly cause serious confusion if multiple scanners are connected to a computer. Are the scanners supposed to be constantly turned on? Won't this cause serious light stress to colonies? Zackrisson et al., 2016 implemented a scanner power managing system to switch of scanners when not scanning while still keeping control over time over which scans are taken by which scanner. *Pyphe* not handling multiple scanners or imposing light stress on colonies would either drastically reduce throughput or accuracy and increase costs (for computers)?

G) The supplementary text does not describe the operations performed or their sequencing in sufficient detail for me to confidently evaluate the analysis process, or to reconstruct what has been done. I kindly ask for more detailed information.

H) The authors argue that the lower precision when using a single time point scan can be compensated for by a higher number of replicates. However, increasing the number of replicates involves very substantial extra manual work and additional costs? E.g. in terms of plates to be poured, scanners and chemicals required, experiments to be started etc? In contrast, measuring multiple times on the same colony is not really a major cost at all? If this is supposed to be a major argument against doing time series, I think the authors should show some empirical support for their point of view.

3) Given the number of pipelines already available:

A) Outside of the viability staining (which I really would love for the authors to succeed in developing), I am not sure what is really innovative or original with *Pyphe*.

B) Moreover, and in addition to the shortcomings mentioned above, the authors seem to not have implemented the major advancement described in the Zackrisson et al., 2016 paper: an accurate conversion of pixel intensities into cell counts. To my mind, this is a serious shortcoming, because of the non-linearity of transmitted light and true population size. Not accounting for this non-linearity is the same as not diluting dense cell cultures when doing manual OD measurements. The consequence of not accounting for this non-linearity is that detected mid and late stage growth will be much attenuated relative the true population size expansion in these stages. Thus, any effect, of genetic background and/or environment, on this part of the growth curve, risk being mis-quantified or completely overlooked.

C) For claims of capturing fitness to be on a more solid footing, the authors may want to mathematically derive selection coefficients from their fitness estimates (see e.g. Stenberg et al., 2016). This would add novelty to their pipeline.

4) A main conclusion is: "We apply *pyphe* to show that the fitness information contained in late endpoint measurements of colony sizes is similar to maximum growth slopes from time series." But:

A) The conclusion is reached by extrapolation from growth on rich YES medium, where all growth curves are canonical to growth in general. I am not sure this is appropriate. Different growth environments clearly have disparate effects on the growth curve (e.g. Warringer et al., 2008). For this conclusion to be of any value, a large number of environments with very different curve behaviors should be considered. Warringer et al., 2011 reached a more convincing conclusion using *S. cerevisiae* natural strains in hundreds of environments.

B) The authors use colony area for this comparison. I am not sure this is ideal. The horizontal expansion rate of a colony is not necessarily a good proxy for the population size expansion rate, and the ratio between the two is rarely constant over time. Early population size expansion is often predominantly horizontal while later expansion is often vertical. Thus, only considering horizontal expansion may lead to mid or late growth being underestimated? This ties into the non-linearity issue.

C) For meaningful biological interpretations to be made, the authors may want to compare the actual growth yield, i.e. the population size change to the end of growth, to the maximum growth rate. This will require some mathematical operations to ensure that growth has indeed ended at the later timepoints and potentially extending the cultivation time beyond 48h. The authors can then link their finding to major microbiological topics, such as r and k selection theory (see e.g. Wei et al., 2019, where the opposite conclusion to that here reported is reached) and thermodynamic considerations of the rate versus the yield of energy limited microbial growth reactions, e.g. MacLean et al., 2008. The activity of many cellular process changes quite dramatically during the growth curve. For example, our understanding of diauxic shifts, glucose and nitrogen catabolite repression, and the different affinities of alternative nutrient influx systems are hard to reconcile with the conclusion that population size at a single time point can well capture the total biology of a growth curve. The authors may want to discuss this. Ibstedt et al., 2015 showed strong correlations in natural strains, such as the ones here used, between effects on maximum growth and yield. However, they demonstrated that this is due to natural co-selection on these fitness components, rather than pleiotropy. Consequently, the here reported conclusion may not necessarily be valid for e.g. gene deletions, which is the major application envisioned by the authors?

5) The normalization procedure here employed and highlighted is a combination of the Zackrisson et al., 2016 and Baryshnikova et al., 2010 approaches, with a minor modification. I am not sure that it brings anything really novel to the table. Moreover:

A) The authors claim that their approach is superior to the Zackrisson and Baryshnikova approaches but does not empirically show this. The Baryshnikova approach – correcting for row and column based bias – makes sense if the error follows such a row – column wise pattern. This is the case for population size at late time points in rich 2% glucose medium – because nothing but the local glucose content is limiting for growth rates at this time. Thus, edge colonies, which experience less competition for the local glucose, expand faster at later stages. However, as Zackrisson et al., 2016 showed, the maximum slope on a rich medium is taken before the local glucose becomes growth limiting. The associated error at this time point therefore does not well follow a row-column wise pattern. Moreover, in harsher environments than 2% glucose YES (i.e. most other environments), where growth at all stages of the growth curve may be limited by other factors than local competition for limited glucose, the same reasoning applies (also to some extent shown by Zackrisson et al., 2016). I think the authors should show error distributions across a plate, in many environments, before and after the Zackrisson, Baryshnikova and *Pyphe* normalization. Moreover they should show that *Pyphe* normalization, across many environments and timepoints, results in more favorable ROC curves. Now it is not clear that the *Pyphe* normalization approach represents an advancement.

B) The authors do not show the effect of normalization on the growth curves, which I think should be done, e.g. in Figure 2A. Moreover, growth curves should be shown on a log-scale such that the exponential phase can be readily distinguished.

C) From a more general perspective: the whole of Figure 2 is based on a single, ideal environment (2% glucose YES) with canonical growth curves, but the conclusions are generalized to the method as such. I am not convinced that this is sufficient. I believe that it is fair to ask for expansion such that also a broad range of non-canonical growth curves are covered – preferentially from environments where growth is limited by other factors than the local glucose concentration. For comparison, Zackrisson et al., 2016 considered six different environments.

6) I really, really appreciate the intentions of the authors in trying to extend their set-up to viability staining, which has the potential to count dead cells and resolve birth and death rates. However, I am not convinced that they yet have succeeded in their intentions.

A) The authors do not really consider, or show, deaths over time and does not estimate death rates. Illuminating how death rates changes as a function of growth and environment, or really just highlighting how often it is substantially above zero at different parts of the growth curve, could advance the field substantially, as non-negligible death rates have confounding effects on key microbiological properties (e.g. Frenoy et al., 2018). A time resolved view on death would much improve the paper and we need to see it.

B) None of the method evaluation that is done for growth (Figure 2) is repeated for the viability staining. The reader has no real clue what precision and accuracy looks like over time, how errors are distributed across and within plates, in different environments, at different time points, how well the normalization works, how false positive rates compare to false negative rates etc. Since this is a method paper, I think this is an essential component.

C) From Figure 4D, E, it seems there is a huge variation in the registered colour intensity that is unrelated to whether cells are dead and alive. For much of the dynamic range, the registered redness seems to only reflect noise? And the fraction of live cells in a colony, in this span, has no real quantitative interpretation. The only reliable distinction seems to be the qualitative separation of colonies with many and few dead cells. This drastically reduces the usefulness of the method?

D) If I understand Figure 4C-E right, there seems to be a strong confounding effect of lysed cells when considering *Pyphe* colony redness: both lysed and alive cells reduce the redness. Hence, colonies with a high fraction of alive cells and colonies with a high fraction of lysed cells can show similar redness? In harsh environments, where a high fraction of cells are first killed and then lysed I imagine that results therefore will be very hard to interpret? This must be a serious shortcoming as compared to e.g. flow cytometry where lysed and alive cells seems to be well separated?

E) The authors’ general conclusion, that there is no overlap between the fraction of dead cells and colony growth, is conceptually very troubling. How can this not be the case, if they really capture population size growth and dead cells respectively? Surely, dead cells do not reproduce: growth, as a consequence, must slow? Or? For example, from Figure 3A, it seems that many colonies grow at a normal rate (i.e. reach an intermediate size), even though only 25% of cells are alive (i.e. the colony redness is 1.25). If, as stated, the detected death is completely disconnected from the detected population size growth, something is fundamentally strange with the detection.

F) From Figure 3B it is clear that there often is a growth impact of phloxine B and that it depends heavily on the environment. One also wonders how genotype-dependent the impact of phloxine B on growth is? If phloxine B has a large impact on growth in many environments and on many genotypes – isn't there a serious risk for confounding effects when measuring both in parallel?

For your reference, we are also providing the individual reviews below; it may be worthwhile to also have a look at these and regard them as suggestions that could help to further improve the paper.

Reviewer #1:

This resources paper describes a modular Python pipeline for automated analysis of microbial colony growth and viability (color). The pipeline integrates basic image analysis, correction and statistics.

Whereas many different research teams have independently developed similar pipelines, it is definitely useful to make a standardized and somewhat user-friendly pipeline available to the broad community, especially for those colleagues lacking the expertise to develop similar pipelines. In addition, if *Pyphe* becomes a success, it could help standardize colony image analysis, and, by extension, fitness data.

The authors mention position effects, but it is unclear to me how they really correct for these. The text mentions "... i.e. cells positioned next to slow growers have better access to nutrients. Indeed, after reference grid normalisation, we often observed a (generally weak but detectable) secondary edge effect for colonies positioned in the next inward row/column (Figure 1—figure supplement 2B right). We found however, that this effect can easily be remedied by an additional row/column median normalization". How can normalization over a complete row or column remove the effects of neighboring cells, especially since the number of fast or slow growers can be very different across different rows and columns? Or are rows/columns at the plate's extremities compared to the inner colonies? Probably very simple, but please explain more clearly.

Instead of using the term "corrected growth rate" values to refer to fitness/growth relative to the WT, perhaps it is better to call this "relative growth rate". When I read "corrected", I am thinking of the removal of non-biological noise such as positional effects.

Figure 2: To me, the color scale is misleading. When I first looked at it, I interpreted the dark red color as being high, only to realize that this is in fact low. Consider re-coloring (e.g. blue-red is color-blindness-friendly, with blue intuitively meaning low).

Reviewer #2:

Kamrad et al. introduces a data acquisition and analysis pipeline for high-throughput microbial growth data: *Pyphe*. They use *Pyphe* to analyze colony growth, and the death component of growth, using moderate scale *S. pombe* experiments. Microbial growth is a central phenotype in microbiology; if correctly measured it can be used as a proxy for fitness. I appreciate the efforts and intentions by the authors, but there are a quite substantial number of similar pipelines available. The authors base their approach on the Zackrisson et al., 2016 pipeline, incorporating some concepts from Wagih and Parts, 2014 and Baryshnikova et al., 2010. But they introduce few novel developments. Moreover, in several critical respects the *Pyphe* pipeline seems like a step backwards. I have some technical and conceptual concerns with the pipeline and I am not sure that the authors yet have benchmarked and evaluated their pipeline to a sufficient extent. The main conclusion highlighted, a correlation between the maximum growth rate and late stage yield data, is not convincing or well illuminated and have been reported before, using a much broader empirical basis. The most exciting part of this paper, the ambition of which I much appreciate, is the expansion of the growth platform to also incorporate viability staining to measure cell death. However, I am not sure that the method is yet put on a sufficiently sound empirical footing. There are question marks concerning what is actually captured, whether the method achieves more than a qualitative resolution and to what extent the staining as such impacts on the growth of cells. Moreover, the main conclusion from this section, that the fraction of dead cells is disconnected from colony growth, is conceptually quite troubling and hints at underlying serious issues with the method. While I am very positively disposed towards the intentions of the authors, I am afraid that quite substantial work remains before *Pyphe* can be regarded as a robust and innovative data analysis pipeline.

1) As an expert user, I have issues with core aspects of the method.

A) The v800 model now seems out of production. For the *Pyphe*-*scan* method to be useful for other labs, clearly it must work with other scanners. At the very least, the authors should show that the pipeline works and provides equivalent data with the 850 model.

B) In this context, it seems unfortunate that authors have chosen not to implement the pixel calibration function of the Zackrisson et al., 2016 approach. This uses a grey-scale calibration strip attached to each fixture to ensure that registered pixel intensities becomes comparable across types of instruments, scanners, external conditions and time – the latter is quite important as the properties of light sources and light receptors change as a function of age.

C) The authors have chosen to, as a default, use a lossy, irreversible compression format for their images, JPG. This format uses very inexact approximations and discards much data when representing the original content of the image. JPG is not recommended when quantitative data is to be extracted from an image. The use of e.g. TIFF is much to be preferred: if the authors want to maintain the JPG format, a minimum requirement is that they show that colony delineations, pixel intensity and background intensity are not affected by the lossy compression format. Otherwise, it should be discarded and no data based on it included.

D) Quality control, QC, is normally a substantial investment of time. It was not clear from the paper how the QC or the QC interface works. If an automated, or at least a semi-automated process is not implemented this is a serious shortcoming and would be another step backwards from the Zackrisson et al., 2016. The importance of a high throughput QC becomes clear when studying the example growth curves supplied together with the software: it is apparent that a large fraction of the slopes extracted are wildly inaccurate, or indeed pure noise, and does not reflect the actual growth. How is this handled? If the user has to filter these out manually, with no software guidance, the pipeline is no longer high-throughput.

E) The graphical interface provides user-friendliness. However, the GUI seems to only be applicable to the later stages of the pipeline? The initial steps are command-line based, which clearly does not square with claims of user-friendliness in general. As an example: to position the grid, the user is requested to input pixel coordinates, as integers, from the image in the command-line interface. This lack of a GUI for the early steps is another step backwards from the graphical interface of plate position or automatic grid detection implemented in Zackrisson et al., 2016?

F) How does *Pyphe* deal with the identification of scanners, when multiple scanners are connected to a computer? Explicitly: how does *Pyphe* in these conditions ensure that the images comes from the scanner that the user thinks the image comes from? In the CLI interface the scanner number can be input, but it is stated that this may change if the scanner is turned off. Such a change will supposedly cause serious confusion if multiple scanners are connected to a computer. Are the scanners supposed to be constantly turned on? Won't this cause serious light stress to colonies? Zackrisson et al., 2016 implemented a scanner power managing system to switch of scanners when not scanning while still keeping control over time over which scans are taken by which scanner. *Pyphe* not handling multiple scanners or imposing light stress on colonies would either drastically reduce throughput or accuracy and increase costs (for computers)?

G) The supplementary text does not describe the operations performed or their sequencing in sufficient detail for me to confidently evaluate the analysis process, or to reconstruct what has been done. I kindly ask for more detailed information.

H) The authors argue that the lower precision when using a single time point scan can be compensated for by a higher number of replicates. However, increasing the number of replicates involves very substantial extra manual work and additional costs? E.g. in terms of plates to be poured, scanners and chemicals required, experiments to be started etc? In contrast, measuring multiple times on the same colony is not really a major cost at all? If this is supposed to be a major argument against doing time series, I think the authors should show some empirical support for their point of view.

2) Given the number of pipelines already available:

A) Outside of the viability staining (which I really would love for the authors to succeed in developing), I am not sure what is really innovative or original with *Pyphe*.

B) Moreover, and in addition to the shortcomings mentioned above, the authors seem to not have implemented the major advancement described in the Zackrisson et al., 2016 paper: an accurate conversion of pixel intensities into cell counts. To my mind, this is a serious shortcoming, because of the non-linearity of transmitted light and true population size. Not accounting for this non-linearity is the same as not diluting dense cell cultures when doing manual OD measurements. The consequence of not accounting for this non-linearity is that detected mid and late stage growth will be much attenuated relative the true population size expansion in these stages. Thus, any effect, of genetic background and/or environment, on this part of the growth curve, risk being mis-quantified or completely overlooked.

C) For claims of capturing fitness to be on a more solid footing, the authors may want to mathematically derive selection coefficients from their fitness estimates (see e.g. Stenberg et al., 2016). This would add novelty to their pipeline.

3) A main conclusion is: "We apply *pyphe* to show that the fitness information contained in late endpoint measurements of colony sizes is similar to maximum growth slopes from time series." But:

A) The conclusion is reached by extrapolation from growth on rich YES medium, where all growth curves are canonical to growth in general. I am not sure this is appropriate. Different growth environments clearly have disparate effects on the growth curve (e.g. Warringer et al., 2008). For this conclusion to be of any value, a large number of environments with very different curve behaviors should be considered. Warringer et al., 2011 reached a more convincing conclusion using *S. cerevisiae* natural strains in hundreds of environments.

B) The authors use colony area for this comparison. I am not sure this is ideal. The horizontal expansion rate of a colony is not necessarily a good proxy for the population size expansion rate, and the ratio between the two is rarely constant over time. Early population size expansion is often predominantly horizontal while later expansion is often vertical. Thus, only considering horizontal expansion may lead to mid or late growth being underestimated? This ties into the non-linearity issue.

C) For meaningful biological interpretations to be made, the authors may want to compare the actual growth yield, i.e. the population size change to the end of growth, to the maximum growth rate. This will require some mathematical operations to ensure that growth has indeed ended at the later timepoints and potentially extending the cultivation time beyond 48h. The authors can then link their finding to major microbiological topics, such as r and k selection theory (see e.g. Wei et al., 2019, where the opposite conclusion to that here reported is reached) and thermodynamic considerations of the rate versus the yield of energy limited microbial growth reactions, e.g. MacLean et al., 2008. The activity of many cellular process changes quite dramatically during the growth curve. For example, our understanding of diauxic shifts, glucose and nitrogen catabolite repression, and the different affinities of alternative nutrient influx systems are hard to reconcile with the conclusion that population size at a single time point can well capture the total biology of a growth curve. The authors may want to discuss this. Ibstedt et al., 2015 showed strong correlations in natural strains, such as the ones here used, between effects on maximum growth and yield. However, they demonstrated that this is due to natural co-selection on these fitness components, rather than pleiotropy. Consequently, the here reported conclusion may not necessarily be valid for e.g. gene deletions, which is the major application envisioned by the authors?

4) The normalization procedure here employed and highlighted is a combination of the Zackrisson et al., 2016 and Baryshnikova et al., 2010 approaches, with a minor modification. I am not sure that it brings anything really novel to the table. Moreover:

A) The authors claim that their approach is superior to the Zackrisson and Baryshnikova approaches but does not empirically show this. The Baryshnikova approach – correcting for row and column based bias – makes sense if the error follows such a row – column wise pattern. This is the case for population size at late time points in rich 2% glucose medium – because nothing but the local glucose content is limiting for growth rates at this time. Thus, edge colonies, which experience less competition for the local glucose, expand faster at later stages. However, as Zackrisson et al., 2016 showed, the maximum slope on a rich medium is taken before the local glucose becomes growth limiting. The associated error at this time point therefore does not well follow a row-column wise pattern. Moreover, in harsher environments than 2% glucose YES (i.e. most other environments), where growth at all stages of the growth curve may be limited by other factors than local competition for limited glucose , the same reasoning applies (also to some extent shown by Zackrisson et al., 2016). I think the authors should show error distributions across a plate, in many environments, before and after the Zackrisson, Baryshnikova and *Pyphe* normalization. Moreover they should show that *Pyphe* normalization, across many environments and timepoints, results in more favorable ROC curves. Now it is not clear that the *Pyphe* normalization approach represents an advancement.

B) The authors do not show the effect of normalization on the growth curves, which I think should be done, e.g. in Figure 2A. Moreover, growth curves should be shown on a log-scale such that the exponential phase can be readily distinguished.

C) From a more general perspective: the whole of Figure 2 is based on a single, ideal environment (2% glucose YES) with canonical growth curves, but the conclusions are generalized to the method as such. I am not convinced that this is sufficient. I believe that it is fair to ask for expansion such that also a broad range of non-canonical growth curves are covered – preferentially from environments where growth is limited by other factors than the local glucose concentration. For comparison, Zackrisson et al., 2016 considered six different environments.

5) I really, really appreciate the intentions of the authors in trying to extend their set-up to viability staining, which has the potential to count dead cells and resolve birth and death rates. However, I am not convinced that they yet have succeeded in their intentions.

A) The authors do not really consider, or show, deaths over time and does not estimate death rates. Illuminating how death rates changes as a function of growth and environment, or really just highlighting how often it is substantially above zero at different parts of the growth curve, could advance the field substantially, as non-negligible death rates have confounding effects on key microbiological properties (e.g. Frenoy et al., 2018). A time resolved view on death would much improve the paper and we need to see it.

B) None of the method evaluation that is done for growth (Figure 2) is repeated for the viability staining. The reader has no real clue what precision and accuracy looks like over time, how errors are distributed across and within plates, in different environments, at different time points, how well the normalization works, how false positive rates compare to false negative rates etc. Since this is a method paper, I think this is an essential component.

C) From Figure 4D, E, it seems there is a huge variation in the registered colour intensity that is unrelated to whether cells are dead and alive. For much of the dynamic range, the registered redness seems to only reflect noise? And the fraction of live cells in a colony, in this span, has no real quantitative interpretation. The only reliable distinction seems to be the qualitative separation of colonies with many and few dead cells. This drastically reduces the usefulness of the method?

D) If I understand Figure 4C-E right, there seems to be a strong confounding effect of lysed cells when considering *Pyphe* colony redness: both lysed and alive cells reduce the redness. Hence, colonies with a high fraction of alive cells and colonies with a high fraction of lysed cells can show similar redness? In harsh environments, where a high fraction of cells are first killed and then lysed I imagine that results therefore will be very hard to interpret? This must be a serious shortcoming as compared to e.g. flow cytometry where lysed and alive cells seems to be well separated?

E) The authors general conclusion, that there is no overlap between the fraction of dead cells and colony growth, is conceptually very troubling. How can this not be the case, if they really capture population size growth and dead cells respectively? Surely, dead cells do not reproduce: growth, as a consequence, must slow? Or? For example, from Figure 3A, it seems that many colonies grow at a normal rate (i.e. reach an intermediate size), even though only 25% of cells are alive (i.e. the colony redness is 1.25). If, as stated, the detected death is completely disconnected from the detected population size growth, something is fundamentally strange with the detection.

F) From Figure 3B it is clear that there often is a growth impact of phloxine B and that it depends heavily on the environment. One also wonders how genotype-dependent the impact of phloxine B on growth is? If phloxine B has a large impact on growth in many environments and on many genotypes – isn't there a serious risk for confounding effects when measuring both in parallel?

Reviewer #3:

The current manuscript describes a comprehensive pipeline that is a wrapper around already available tools and implements already described approaches (e.g. grid normalisation) and which can be used to analyse imaging data collected using flatbed scanners for high-throughput fitness screens. While the paper is very clear and well written and the code deposited in a public repository appears to be well crafted and documented, I am unsure there is enough novelty in this tool or in the experimental validation reported in the current manuscript to be of interest for the general readership of *eLife*. If I understand this correctly, there aren't any critical steps implemented in this pipeline which had not been reported or implemented before, which makes me think that a journal where this kind of tools are reported might be a better home for the current manuscript. Unfortunately I lack the expertise in the specific area of high-throughput phenotypic screens to be able to judge whether the substantial work presented here constitutes a technical improvement and a practical tool that might be widely used or rather a step change in the field. Without more competing arguments in favour of the latter and without a clear indication of a novel approach rather than implementation of already described tools and techniques I cannot fully support this manuscript for publication in *eLife*.

[Editors' note: further revisions were suggested prior to acceptance, as described below.]

Thank you for resubmitting your work entitled "*Pyphe*, a python toolbox for assessing microbial growth and cell viability in high-throughput colony screens" for further consideration by *eLife*. Your revised article has been evaluated by Aleksandra Walczak as the Senior Editor and a Reviewing Editor.

The manuscript has been improved but there are some remaining issues that need to be addressed before acceptance, as outlined below:

All reviewers agree that you and your co-authors have responded adequately to the concerns that were raised and that the paper has matured significantly. That said, we would still recommend addressing two specific points in a bit more detail in the paper, so that the readers are at the very least made aware that these might be potential concerns. Firstly, we think it is important for growth measures to be as good proxies for population size as possible. Accounting for the non-linearity of optical and cell density is important, even if this is often ignored (because the difference is not huge, or because of technical issues). Second, there still is some concern about the lack of correlation between measured death and the measured fitness proxy – one reviewer is not sure that this is not due to one or both measures being afflicted by a large error. We understand that this is not easily solved, but believe it is fair to mention it explicitly in the paper as a potential concern.

This resources paper describes a modular Python pipeline for automated analysis of microbial colony growth and viability (color). The pipeline integrates basic image analysis, correction and statistics. The pipeline will be a useful tool for the broad community and may help standardize high-throughput automated growth measurements for microbes.

---

## [Author Response]

Reviewer #1:[…]The authors mention position effects, but it is unclear to me how they really correct for these. The text mentions "... i.e. cells positioned next to slow growers have better access to nutrients. Indeed, after reference grid normalisation, we often observed a (generally weak but detectable) secondary edge effect for colonies positioned in the next inward row/column (Figure 1—figure supplement 2B right). We found however, that this effect can easily be remedied by an additional row/column median normalization". How can normalization over a complete row or column remove the effects of neighboring cells, especially since the number of fast or slow growers can be very different across different rows and columns? Or are rows/columns at the plate's extremities compared to the inner colonies? Probably very simple, but please explain more clearly.

Thank you for highlighting the need for additional explanation and more clarity here. All arrayed yeast colonies will show differences in growth based on where they are located in the plate. This is partly due to the classical edge effect reflecting that colonies on the edge are exposed to a larger volume of agar without competition/detoxification effects from neighbours. This affects entire rows and columns largely uniformly so could be corrected with a simple row/column median normalisation. But in addition to the edge effect, there are other spatial effects with no defined patterns. These present themselves in the form of growth differences in different areas of the plate, without a clear pattern or size, and originate from uneven plate pouring, uneven heating or drying of the plate, uneven distribution of nutrients/toxins in the agar or pinning errors. Both these effects can efficiently be corrected using a reference grid normalisation which essentially compares the size of any given colony to those of wild type control colonies arrayed around it. We have expanded Appendix 2 with additional details and explanations.

“Or are rows/columns at the plate's extremities compared to the inner colonies?”

Yes. While the grid correction is very good at removing most technical noise (it typically reduces the CV by 4-fold), it can potentially introduce artefacts in the form of negative fitness values and a secondary edge effect. In the case of the secondary edge, this is because the edge effect only affects the outermost rows/columns. But in order to compute the relative fitness, colonies in the second row/column are compared to neighbouring colonies, which includes colonies on the edge (which increases the value that is being compared to). Since this affects entire rows/columns uniformly, it can be corrected by dividing by row/column medians.

“How can normalization over a complete row or column remove the effects of neighboring cells, especially since the number of fast or slow growers can be very different across different rows and columns?”

As pointed out, a median correction is not valid if the median is not a good estimate of the null effect. For this reason, we strongly discourage row/column median normalisation for plates in 96-colony format (where the median is computed from only 8 or 12 values). If plates contain a large number of slow or fast growers, a median normalisation is also unsuitable as pointed out, especially if these are distributed non-randomly in the plate. For work with knockout libraries, where most gene knock-outs have no effect in any given condition, the additional row/column median normalisation effectively neutralises the secondary edge effect. Being a toolbox, *pyphe* requires the user to think about their experimental design, choice of control strains, and plate layout to choose a suitable normalisation strategy from the options provided. Although this will require some extra time from the user, we believe that this level of understanding is necessary in order to obtain reliable, interpretable results. This flexibility will allow researchers to tailor data analysis to their experiment and will foster the uptake of *pyphe* in diverse labs and settings. We have improved and expanded Appendix 2, following on from the above arguments.

Instead of using the term "corrected growth rate" values to refer to fitness/growth relative to the WT, perhaps it is better to call this "relative growth rate". When I read "corrected", I am thinking of the removal of non-biological noise such as positional effects.

The grid correction takes care of three things simultaneously: (1) it converts colony size into an easily interpretable value by reporting a ratio relative to WT; (2) it makes results comparable across different plates/batches as long as the same WT strain is used used and grown in the same way; and (3) it corrects for within-plate positional effects which become apparent due to the same WT grid strain showing different fitness in different plate positions. We agree that this reflects the nature of the normalisation better and have followed your suggestion. We now explicitly point out the relative nature of the corrected fitness score (subsection “*Pyphe* enables analysis pipelines for fitness-screen data“) and adapted the term “relative growth rate” in several places in the manuscript.

Figure 2: To me, the color scale is misleading. When I first looked at it, I interpreted the dark red color as being high, only to realize that this is in fact low. Consider re-coloring (e.g. blue-red is color-blindness-friendly, with blue intuitively meaning low).

Thank you for this suggestion. We agree that an inversion of the colour scale is more intuitive and have changed heatmaps. We have opted against using a two-colour scheme which would imply a neutral mid-point or inflection point not present in correlation data.

Reviewer #2:Kamrad et al. introduces a data acquisition and analysis pipeline for high-throughput microbial growth data: Pyphe. […] While I am very positively disposed towards the intentions of the authors, I am afraid that quite substantial work remains before Pyphe can be regarded as a robust and innovative data analysis pipeline.

Before the point-by-point discussion, we would like to describe some fundamental differences between scan-o-matics and *pyphe* to demonstrate that *pyphe* is not a step backwards but a unique, novel approach with a distinct philosophy and streamlined implementation. Work on this project has evolved for almost four years and the initial goal was not to develop a new software solution. Indeed, it is a guiding philosophy of our labs to not reinvent the wheel, but to develop new tools only when they are needed and useful for the community. We have summarised key differences between scan-o-matics and *pyphe* in Author response table 1. These have been motivated by: (1) users conducting screens of different sizes (ranging from 3 to 3000 plates) with different questions and methods; (2) the need for users to have full control and insight into different steps of the analysis pipeline; (3) limited space in temperature-controlled rooms and incubators; (4) the need for simplified hardware set-ups not requiring scanner modifications (safety and warranty issues), dedicated LAN networks or power switches.

Author response table 1

Our preference for using endpoints and redness estimates as fitness inputs was a result, and not an initial expectation. Indeed we were for a long time very skeptical about end-point measurements, but the data of over half a million colony sizes recorded in our labs so far has shown that they can be used as a fitness proxy in a vast majority of cases. As resources are always limiting, we feel these are very important points to report.

For scientific questions and projects for which the priority lies in throughput, the flexible use of different fitness proxies, and streamlined scripted analysis, *pyphe* is a highly efficient and precise workflow that in our hands performs significantly better than its predecessors. There are other applications, however, like those that require the estimation of precise cell numbers in colonies for which *pyphe* is not designed and where scan-o-matics is the method of choice.

1) As an expert user, I have issues with core aspects of the method.A) The v800 model now seems out of production. For the Pyphe-scan method to be useful for other labs, clearly it must work with other scanners. At the very least, the authors should show that the pipeline works and provides equivalent data with the 850 model.

We agree and had in fact already ordered a V850 scanner. The new model is supported by SANE, so we fully expect this to be straightforward. Unfortunately, with the current closure of our institutions due to the pandemic, we are at this time not able to validate it for use with *pyphe* in the lab. We will validate and update *pyphe* and the GitHub documentation for the new scanner model, as soon as we can.

B) In this context, it seems unfortunate that authors have chosen not to implement the pixel calibration function of the Zackrisson et al., 2016 approach. This uses a grey-scale calibration strip attached to each fixture to ensure that registered pixel intensities becomes comparable across types of instruments, scanners, external conditions and time – the latter is quite important as the properties of light sources and light receptors change as a function of age.

We have been working with pixel calibration strips during the development of *pyphe* (the fixture we use and have published cutting vectors for has a space to accommodate it) but have moved away from them for most daily uses (an exception are colony ageing experiments where the same plate is scanned once daily over several weeks). They are somewhat expensive and difficult to procure, and we have found them unnecessary for most of our purposes. A different calibration does not affect the binary classification of colonies and background which is all we need to quantify areas with high precision. We illustrate this in Author response image 1 by transforming the same image with 2 different hypothetical calibration functions and then analysing colony sizes with *pyphe-quantify*. The results are highly consistent, even in the case of the second transformation which is extreme and causes visible distortion of the image.

**Author response image 1. sa2fig1:** Pixel calibration is not required for accurate determination of colony sizes. Top row: calibration functions applied to the original scanned image. The first function is a linear transformation that scales the image to fill the entire 8bit range. We apply this to images in batch (but not timecourse) analysis by default. The other functions are 3rd-degree polynomials (as used by scan-o-matics). Middle row: Transformed images with upper right corner magnified. The third function has strong non-linear components which result in visible distortion of the image. Bottom row: colony sizes obtained with *pyphe-quantify batch* of the transformed images versus the original image. The median of the relative error abs(size(transformed)-size(original))/size(original) across the plate is noted. This is negligibly small when compared to the variation of colony sizes.

For virtually all applications of *pyphe*, we compare colony areas or redness scores within the same plate to produce corrected fitness values relative to the control strain. Differences between different scanners or the same scanner over a long time will thereby be corrected as they affect all colonies on a plate equally.

Special consideration needs to be given to growth curves, where uncorrected information from multiple images of the same plate is used. As described in Appendix 2, *pyphe-quantify timecourse* identifies colony positions in the last plate of the timecourse and then applies this mask to all background-subtracted images, each time summing up the intensities of all pixels in the masked areas. Zackrisson et al., 2016 noticed some variation in scanner calibration between consecutive scans (Supplementary Figure 3D). Somewhat surprisingly, we have found that our scans obtained with the V800 model are consistent enough over a period of a few days (scanners are undisturbed, light-protected and scanner age changes over much larger time-scales) to produce smooth growth curves (Author response image 2). We suspect that part of the variation/noise in the growth curves observed by Zackrisson et al., 2016 comes from the hard reboot of thecaner between every image, whichh we do not do (see also point 1F).

**Author response image 2. sa2fig2:** Examples of raw growth curves obtained with *pyphe* setup. Shown are 12 growth curves from the first row of a 1536 plate of 57 *S. pombe* wild strains (same data as Figure 2 in manuscript) analysed with *pyphe-quantify* in timecourse mode.

Fitting of lines to determine the maximum slope is an additional step to compensate for noise in the data (more noisy data can be compensated by fitting over more timepoints, and the user has the option to do so). After grid normalisation and reporting fitness results relative to the control strain, data are again comparable across days, instruments etc.

C) The authors have chosen to, as a default, use a lossy, irreversible compression format for their images, JPG. This format uses very inexact approximations and discards much data when representing the original content of the image. JPG is not recommended when quantitative data is to be extracted from an image. The use of e.g. TIFF is much to be preferred: if the authors want to maintain the JPG format, a minimum requirement is that they show that colony delineations, pixel intensity and background intensity are not affected by the lossy compression format. Otherwise, it should be discarded and no data based on it included.

Thank you for raising this point. Our choice of image format was not by accident but a carefully considered trade-off to reduce storage space requirements. A tiff image of a single plate scanned at 600dpi is approximately 4MB large, whereas the corresponding jpg (converted using ImageMagick’s default parameters) is usually only in the range of 180-580KB, reducing image storage needs by a factor of ~20. This is relevant, as *pyphe* is designed specifically for high-throughput pipelines.

In order to address your concern, we have re-analysed images from the 57 *S. pombe* wild strain growth curve experiment on rich medium (Figure 2). This experiment contained 145 images of the same plate, scanned every 20 minutes. We have analysed each image separately in tiff and jpg format, both with *gitter* and with *pyphe-quantify batch*. We have computed the Pearson correlation between results obtained with both image formats and achieved an overall correlation of 0.999976 for analysis with *gitter* and 0.999964 with *pyphe-quantify batch*. We have also computed the relative error introduced by conversion to jpg, defined as abs(size(jpg)size(tiff_image))/size(tiff), which has a median of 0.00245 for *gitter* and 0.00333 for *pyphe*. We have computed both of these measures plate-wise (Author response image 3) and find that the error introduced by conversion is consistently low across the growth curve when considering that early images with smaller, fainter colonies are harder to analyse. These relative errors need to be put into perspective by comparison of the biological signal (here estimated by the median absolute deviation of all colonies in each plate). The error introduced by conversion is negligible compared to the biological signal.

**Author response image 3. sa2fig3:** Image conversion to jpg has negligible impact on results. Each image of a growth curve consisting of 145 images (shown on x-axis) was analysed in the original tiff format and in the converted jpg, using *gitter* (right) and *pyphe-quantify* in batch mode (left). The correlation (blue line) is extremely high for all images (>0.996) and increases as colonies get larger and darker. The median relative error abs(size(jpg)-size(tiff_image))/size(tiff) is shown (orange) and is practically 0 compared to the biological signal (median absolute deviation of all colony sizes per plate).

We also confirmed that image conversion makes no difference in the case of growth curves, where *pyphe* reports the sum of pixel intensities in the footprint of the colony in the final image. Using *pyphe-quantify* in timecourse mode on the entire image series in jpg and tiff format produces an overall Pearson correlation of 0.999998 with a median relative error of 0.00087 across all colonies and timepoints.

However, we fully accept that some researchers may prefer to work with lossless image formats and now offer the option in *pyphe-scan* to produce images in tiff. *Pyphe-quantify* is already flexible with regards to image format. Please note that *pyphe-scan*, in any case, saves the original scans in tiff format (which we usually delete or archive once image processing is complete, depending on the project).

D) Quality control, QC, is normally a substantial investment of time. It was not clear from the paper how the QC or the QC interface works. If an automated, or at least a semi-automated process is not implemented this is a serious shortcoming and would be another step backwards from the Zackrisson et al., 2016. The importance of a high throughput QC becomes clear when studying the example growth curves supplied together with the software: it is apparent that a large fraction of the slopes extracted are wildly inaccurate, or indeed pure noise, and does not reflect the actual growth. How is this handled? If the user has to filter these out manually, with no software guidance, the pipeline is no longer high-throughput.

*Pyphe-growthcurves*, our tool for growth curve analysis is written to be as flexible as possible, and we use it for liquid growth curve analysis as well. The example data supplied is from liquid cultures from a plate reader experiment and indeed of much poorer quality than what we would expect from solid growth curves. We have now provided more appropriate example data (from the 57 wild strain experiment, Figure 2). We have additionally improved the tool to perform some automated growth curve QC and tag spurious curves in a new column of the output file if the R^2^ of the fitted line is <0.95 or a significantly negative slope is detected in the growth curve (this happens quite frequently for plate reader growth curves). We consciously avoid fitting parametric growth models to colony area data and believe this should be left to expert users, if required, who can easily access *pyphe-growthcurve* data thanks to the use of standardised, simple, human-readable intermediate steps.

For endpoint measurements, we recommend a QC strategy based on colony circularities which is clearly described in the manuscript. We have now implemented these two key steps in *pyphe-interpret* and updated the documentation, so no manual QC is required by the user for typical endpoint experiments.

E) The graphical interface provides user-friendliness. However, the GUI seems to only be applicable to the later stages of the pipeline? The initial steps are command-line based, which clearly does not square with claims of user-friendliness in general. As an example: to position the grid, the user is requested to input pixel coordinates, as integers, from the image in the command-line interface. This lack of a GUI for the early steps is another step backwards from the graphical interface of plate position or automatic grid detection implemented in Zackrisson et al., 2016?

Thank you for raising this point. We agree that this had not been documented very well in the previous manuscript. We initially have not implemented an automatic grid correction since all our plates were scanned with the same fixture taped to the scanners, so colony positions were highly consistent across images. Users actually do not need to input pixel coordinates if they are using the fixture and scanning parameters provided by us (this is done using the *--grid pp_384* or *--grid pp_1536* option). This information has been updated. Simultaneously, with the goal of maximum flexibility, *pyphe-quantify* offers the option of manually defining grid positions. Getting those coordinates is trivial and can be done, for example, in Microsoft Paint by hovering the cursor over the colony (we have now pointed this out in the tool manual). The option to manually define grid positions is important in our experience, as automatic gridding is the step where most image analysis tools typically fail (especially if plates have many missing colonies, images are rotated or otherwise of low quality). However, we fully agree that manual entering of colony coordinates is awkward and have now implemented automatic grid detection functionality. It is based on detecting peaks in image pixel rows/columns and is used by setting the *--grid* argument to *auto_96*, *auto_384* or *auto_1536*.

Secondly, we do not agree with the claim that GUIs are more user-friendly in general. They can be useful in many instances and, as pointed out, we do provide a GUI for one of our tools but have otherwise moved away from this for three reasons. First, GUIs struggle with cross-platform compatibility (esp. without browser-based implementation) and are time-consuming to build. Second, GUIs only really make sense if they have interactive/dynamic components, which our tools don’t require. Using a pipeline for data analysis, where tools with simple, well-defined purposes operate on human-readable files, is in our opinion a preferable solution to integrating all functionality in a complicated GUI with tabs, menus and submenus. All *pyphe* tools require only a small set of parameters to start the analysis, which then proceeds without user-input. The well-documented command-line interfaces follow the same scheme and are straightforward to use without any knowledge of computer programming. They are based on the powerful argparse package which checks user inputs carefully. Third, using command line calls allows our pipeline to be scriptable. It is therefore easy to document its use exactly, reproduce results and re-run analyses quickly if the input data has changed/expanded.

F) How does Pyphe deal with the identification of scanners, when multiple scanners are connected to a computer? Explicitly: how does Pyphe in these conditions ensure that the images comes from the scanner that the user thinks the image comes from? In the CLI interface the scanner number can be input, but it is stated that this may change if the scanner is turned off. Such a change will supposedly cause serious confusion if multiple scanners are connected to a computer. Are the scanners supposed to be constantly turned on? Won't this cause serious light stress to colonies? Zackrisson et al., 2016 implemented a scanner power managing system to switch of scanners when not scanning while still keeping control over time over which scans are taken by which scanner. Pyphe not handling multiple scanners or imposing light stress on colonies would either drastically reduce throughput or accuracy and increase costs (for computers)?

The implementation done by scan-o-matics is a clever and well-written solution to both the backlight and scanner identification problem. We have in the beginning implemented the complete setup from Zackrisson et al., 2016 using the LAN power switcher and scanners controlled by the scan-o-matics software interface. We have moved away from this for the following reasons:

1) We have been using only V800 scanners, the replacement model of the V700 used by Zackrisson et al., 2016, and could not reproduce the problem with the light staying on. In our hands, the light switches off promptly after each scan.

2) We suspect that part of the variation/noise in the growth curves observed by Zackrisson et al., 2016 comes from the hard reboot of the scanning between every image, and we do not seem to have these issues (see point 1B).

3) The power switch has not been easy to buy through our procurement system as it is non-standard electrical equipment. It takes time to set up (requiring a dedicated router and IP address configuration) and has not been entirely stable in our hands.

4) Using the power switcher requires a hardware modification of the scanner which many users will be uncomfortable with and which voids the warranty. It was further not compatible with the UK fire safety/electrical safety regulations for us to modify the scanner without obtaining a certificate of the modification. We found nobody that was willing, or legally able, to certify the electrical safety of a scanner modified by ourselves.

5) Scan-o-matics avoids having to turn on two scanners at once by dynamically moving timepoints. This can lead to an uneven spacing of timepoints which complicates downstream analysis.

Setting up an experiment with multiple scanners with *pyphe-scan-timecourse* is straightforward. One simply needs to prepare and connect the first scanner and start scanning with --scanner 1, then connect the second scanner to the computer and start scanning with --scanner 2, etc. This is now more clearly documented in the tool’s help page.

G) The supplementary text does not describe the operations performed or their sequencing in sufficient detail for me to confidently evaluate the analysis process, or to reconstruct what has been done. I kindly ask for more detailed information.

We have expanded Appendix 1 and 2 to describe *pyphe’s* algorithms in more detail. We have now also added Appendix 3 (description of *pyphe-growthcurves*) and Appendix 4 (description of *pyphe-interpret*). Information on how to use each tool may change in future versions and is given in the tools’ inbuilt help (accessible by calling the tool with the -h option) and on GitHub.

H) The authors argue that the lower precision when using a single time point scan can be compensated for by a higher number of replicates. However, increasing the number of replicates involves very substantial extra manual work and additional costs? E.g. in terms of plates to be poured, scanners and chemicals required, experiments to be started etc? In contrast, measuring multiple times on the same colony is not really a major cost at all? If this is supposed to be a major argument against doing time series, I think the authors should show some empirical support for their point of view.

We agree that this point will be stronger with supporting evidence which we now provide. First, we have conducted a power analysis illustrating the trade-off between more replicates and higher measurement precision (Author response image 4). Using a CV of 2% for scan-o-matic, as reported by Zackrisson et al., and a CV of 6%, as reported in our knock-out screen (Figure 2), we have calculated the statistical power (1 – chance of type II error, i.e. non-rejection of a false null hypothesis) dependent on the difference in means of the input phenotypes. We achieve similar power using the number of replicates shown below and note that both methods are able to detect even subtle (10%) growth differences reliably.

**Author response image 4. sa2fig4:** The analysis shows the number of replicates required with scan-o-matics and with *pyphe* in order to achieve the same statistical power.

We further illustrate our response to this point using the example of an experiment we recently conducted, where we measured ~90 non-coding RNA mutants in 9 replicates across ~140 conditions (Rodriguez-Lopez et al., in preparation). This experiment comprises 3 plate layouts per condition (to accommodate all 9 replicates), which amounts to 420 plates in total. Assuming we could have obtained the same statistical power with 3 replicates in scan-o-matics, we have compiled Author response table 2 that breaks down costs and other requirements:

Author response table 2

2) Given the number of pipelines already available:A) Outside of the viability staining (which I really would love for the authors to succeed in developing), I am not sure what is really innovative or original with Pyphe.

Our earlier responses have lined out how we have established a new, common framework for phenotyping analysis. It is important to note that *gitter* (Wagih and Parts, 2014) and *grofit* (Kahm et al., 2010), two popular R packages for image and growth curve analysis, are now archived and no longer installable via install.packages(). So despite the number of publications on the topic, the tools actually available to potential users are very few in practice. Researchers want and need tools which are straightforward to install and use and which fit into their existing workflow and data. *Pyphe* was designed with this in mind and is a unique, comprehensive end-to-end solution for various phenotyping scenarios.

Comparing *pyphe* specifically to scan-o-matics, several points of fundamental difference are highlighted by the reviewer. Moreover, *pyphe* has substantially expanded functionality, being able to process endpoints and growth curves as well as colony sizes and colony redness within the same framework. It further implements downstream statistical analysis and hit calling. *Pyphe* is, to our knowledge, the first platform with such a scope.

Besides *pyphe* itself, this manuscript contains a substantial amount of biological data and new findings and makes wide-ranging conclusions that will be important for everyone working on colony-based screens, regardless of whether they use *pyphe* or not. Briefly, these are (1) the observation that endpoints are highly correlated with growth rate and can be used as a fitness proxy which is much easier to obtain, (2) that colony redness is a reproducible, orthogonal and independent fitness readout easily obtained from the same colony, and (3) that redness scores reflect colony viability.

B) Moreover, and in addition to the shortcomings mentioned above, the authors seem to not have implemented the major advancement described in the Zackrisson et al., 2016 paper: an accurate conversion of pixel intensities into cell counts. To my mind, this is a serious shortcoming, because of the non-linearity of transmitted light and true population size. Not accounting for this non-linearity is the same as not diluting dense cell cultures when doing manual OD measurements. The consequence of not accounting for this non-linearity is that detected mid and late stage growth will be much attenuated relative the true population size expansion in these stages. Thus, any effect, of genetic background and/or environment, on this part of the growth curve, risk being mis-quantified or completely overlooked.

We agree that the measurement of true cell numbers is a distinguishing feature of the scan-o-matics pipeline. As *pyphe* is not meant at all to be a simplified clone of scan-o-matics, it has different feature sets, strengths and weaknesses. We now explicitly state this limitation of *pyphe* in the main text. If high-throughput measurements of true cell numbers should really be required for an experiment, we recommend scan-o-matics to potential users. However, obtaining true cell numbers as implemented in scan-o-matics assumes that the relationship between pixel intensity and cell number does not change between conditions and strains, and it is unclear to which extent this is normally valid.

Furthermore, it complicates the analysis considerably, effectively restricting it to specialist laboratories. But most importantly, we think that true cell numbers are not actually required to answer the vast majority of questions investigated with colony-based screens. The true “fitness” in a natural setting is rarely ever measured but approximated through a linked readout in the laboratory. Colony footprints are an intuitive fitness proxy reflecting how well a strain/colony has performed in the environment. This readout has served biologists incredibly well for decades and is well suited for answering the questions posed in most laboratories. These questions are normally one of the following two types. Type 1 requires a relative quantitative fitness readout and/or classification of “faster/slower growing than another strain in the same experiment”. For example, for GWAS, phenotype vectors are often centred to mean 0, scaled to variance 1 and transformed to normal shape by box-cox or similar before analysis. For this approach, obtaining a readout which is increasing with cell number (even if not in a strictly linear fashion) will result in similar outcomes.

Type 2 concerns profiling approaches for functional genomics and these require a reproducible readout which reflects aspects of physiology. Profile vectors are then used in multivariate analysis, such as clustering, which reveals similarities between genes. This approach requires no mechanistic understanding of what the readout means; it is even blind to the conditions used to obtain them. Instead, they have to be reproducible, precise and measurable in high numbers. Colony sizes are ideally suited to both of these types of questions, supported by the remarkable recent discoveries made using colony-size screens (e.g. Kuzmin et al., 2018 or Galardini et al., 2019).

To validate this conclusion, we would be keen to directly compare the results obtained with *pyphe* to those obtained with the full scan-o-matics setup. To this end, we have tried to analyse the example image data set provided by Zackrisson et al., 2016 (https://github.com/Scan-oMatic/scanomatic/wiki/Example-experimental-data). We have, however, run into difficulties with that data and think it would be best to open a direct dialogue about this issue and whether or how best to pursue the comparison.

C) For claims of capturing fitness to be on a more solid footing, the authors may want to mathematically derive selection coefficients from their fitness estimates (see e.g. Stenberg et al., 2016). This would add novelty to their pipeline.

We could not find this publication. There does not seem to be a publication from this first author in that year. More generally, we do not feel that deriving selection coefficients would be useful at this time, as it is not the core area of expertise of our labs and we have no immediate application for it. However, *pyphe* is set up to become a collaborative project and welcomes code contributions from the community.

3) A main conclusion is: "We apply pyphe to show that the fitness information contained in late endpoint measurements of colony sizes is similar to maximum growth slopes from time series." But:A) The conclusion is reached by extrapolation from growth on rich YES medium, where all growth curves are canonical to growth in general. I am not sure this is appropriate. Different growth environments clearly have disparate effects on the growth curve (e.g. Warringer et al., 2008). For this conclusion to be of any value, a large number of environments with very different curve behaviors should be considered. Warringer et al., 2011 reached a more convincing conclusion using *S. cerevisiae* natural strains in hundreds of environments.

We agree and have now collected a bigger data set for cell growth in 8 additional conditions. These conditions have been specifically designed to produce diverse growth dynamics, using combinations of different carbon sources, salt stress, and different nitrogen sources. We show plots for each individual condition in the new Figure 2—figure supplement 1. We have calculated for each condition the correlation between endpoint and maximum slope of the growth curve (new Figure 2—figure supplement 2) and obtain a medium correlation of 0.95, thus confirming our earlier conclusion.

B) The authors use colony area for this comparison. I am not sure this is ideal. The horizontal expansion rate of a colony is not necessarily a good proxy for the population size expansion rate, and the ratio between the two is rarely constant over time. Early population size expansion is often predominantly horizontal while later expansion is often vertical. Thus, only considering horizontal expansion may lead to mid or late growth being underestimated? This ties into the non-linearity issue.

We agree that these are valid and important theoretical considerations. In fact, we have carefully considered these very same points when we set out to investigate the relationship between endpoints and other growth-curve parameters. Please note that *pyphe-quantify* in timecourse mode reports the sum of pixel intensities, so it does take into account thickness as well as area. We agree with your observations above and with the argument that these could complicate our analysis. Yet, the fact that colony areas do correlate so well with maximum slopes indicates that these points have a comparatively minor impact and can be ignored in practice. By using genetically diverse wild strains for these experiments, we covered strains with highly diverse morphology and growth behaviour. However, such considerations might matter more for other microbial species, and we have added this potential caveat in the main text. *Pyphe-quantify* in batch mode also reports the average pixel intensity as well as the colony area by default (the relationship between which we show in Figure 1—figure supplement 1B), so the user has the option to use those instead.

C) For meaningful biological interpretations to be made, the authors may want to compare the actual growth yield, i.e. the population size change to the end of growth, to the maximum growth rate. This will require some mathematical operations to ensure that growth has indeed ended at the later timepoints and potentially extending the cultivation time beyond 48h. The authors can then link their finding to major microbiological topics, such as r and k selection theory (see e.g. Wei et al., 2019, where the opposite conclusion to that here reported is reached) and thermodynamic considerations of the rate versus the yield of energy limited microbial growth reactions, e.g. MacLean et al., 2008. The activity of many cellular process changes quite dramatically during the growth curve. For example, our understanding of diauxic shifts, glucose and nitrogen catabolite repression, and the different affinities of alternative nutrient influx systems are hard to reconcile with the conclusion that population size at a single time point can well capture the total biology of a growth curve. The authors may want to discuss this. Ibstedt et al., 2015 showed strong correlations in natural strains, such as the ones here used, between effects on maximum growth and yield. However, they demonstrated that this is due to natural co-selection on these fitness components, rather than pleiotropy. Consequently, the here reported conclusion may not necessarily be valid for e.g. gene deletions, which is the major application envisioned by the authors?

Thank you for raising this interesting point. We fully agree with everything stated in principle but do not believe that this is relevant here. Endpoint colony sizes on solid media cannot be used to determine growth yields. Colonies are densely arrayed and keep growing until the agar is depleted of the limiting nutrient. This competition for resources means that rather than each strain having the same amount of nutrients to grow (as would be required to determine yield), each strain has roughly the same amount of time to grow, which means that endpoints largely reflect growth rate, as our analyses show. This is fundamentally different to work in liquid media, as used by (Ibstedt et al., 2015), where each strain grows in its own well/flask without competition from other strains.

“For example, our understanding of diauxic shifts, glucose and nitrogen catabolite repression, and the different affinities of alternative nutrient influx systems are hard to reconcile with the conclusion that population size at a single time point can well capture the total biology of a growth curve.”

We agree that a single data point cannot capture the total biology of a growth curve and we certainly do not claim that it does so. But in the end, most quantitative analyses require a simple numerical input to be extracted from the growth curves. We do show that endpoint colony sizes are an accurate reflection specifically of the maximum slope so these both reflect the key growth-curve parameter. By providing easy access to the raw growth-curve data and plotting all curves as pdf, users can detect unusual growth dynamics and specialised users can easily perform additional, specific analyses.

4) The normalization procedure here employed and highlighted is a combination of the Zackrisson et al., 2016 and Baryshnikova et al., 2010 approaches, with a minor modification. I am not sure that it brings anything really novel to the table. Moreover:A) the authors claim that their approach is superior to the Zackrisson and Baryshnikova approaches but does not empirically show this. The Baryshnikova approach – correcting for row and column based bias – makes sense if the error follows such a row – column wise pattern. This is the case for population size at late time points in rich 2% glucose medium – because nothing but the local glucose content is limiting for growth rates at this time. Thus, edge colonies, which experience less competition for the local glucose, expand faster at later stages. However, as Zackrisson et al., 2016 showed, the maximum slope on a rich medium is taken before the local glucose becomes growth limiting. The associated error at this time point therefore does not well follow a row-column wise pattern. Moreover, in harsher environments than 2% glucose YES (i.e. most other environments), where growth at all stages of the growth curve may be limited by other factors than local competition for limited glucose, the same reasoning applies (also to some extent shown by Zackrisson et al., 2016). I think the authors should show error distributions across a plate, in many environments, before and after the Zackrisson, Baryshnikova and Pyphe normalization. Moreover they should show that Pyphe normalization, across many environments and timepoints, results in more favorable ROC curves. Now it is not clear that the Pyphe normalization approach represents an advancement.

We fully agree with these arguments. A row/column median normalisation only makes sense if (a) the error follows a row/column pattern, and (b) if the error can be estimated reliably. Except for edge effects, errors do not normally follow row/column patterns requiring a different approach. One approach applied previously to combat non-row/column variation is to create a normalisation surface by convolving the image with a median filter. This assumes that most colonies show no response in the condition and creates problems at the edge where the surface is undefined. A normalisation surface based on control strains is a great solution which makes fewer assumptions and makes results intuitively interpretable. Our implementation of the reference grid normalisation is similar in that we use scipy’s interpolate.griddata function with a cubic interpolation. We do not claim that our implementation is better in the sense that it delivers lower noise levels, and we refer frequently to Zackrisson et al., 2016. However, we have made changes to the original implementation which improve data completeness, slightly increase throughput, and facilitate quality control. These are:

1) We recommend placing two 96 grids on each 1536 plate: one in the top left position and one in the bottom right. This leaves 192 more positions per plate to be filled by strains to be assayed.

2) We have implemented a statistical prediction method for predicting colony sizes in the two missing corners of the plate (bottom left and top right), which allows us to extrapolate the grid to cover the entire plate. Together with point (1), this means that we can predict null-effects for the entire plate without loss of data (key for large libraries which would otherwise have to be re-arranged).

3) We recommend including a third 96 grid of control strains (same as the grid or a mix of a few strains for use as positive controls). This enables easy plate-level quality control.

4) We actively look for grid positions where colonies are missing due to pinning errors, throw a warning and set all neighbouring colonies to NA.

During testing of our normalisation strategy, we noticed the secondary edge effect (Figure 1—figure supplement 2D). This is essentially due to colonies in the second row/column being compared to colonies on the edge. This is an error which follows a clear row/column pattern and can easily be corrected with a row/column median normalisation. But this requires that most of the strains in each row/column show no phenotype (as is usually the case when working with knock-out collections, we specifically now point this out in Appendix 2). Generally, *pyphe* gives the user the choice of using both normalisations alone or sequentially (or none at all). We think that showing the requested direct comparison between row/column median normalisation and grid normalisation is not necessary, as the superiority of the latter has been well documented in the previous work by Zackrisson et al., 2016. As mentioned above, we would like to directly compare *pyphe* to scan-o-matics if we can make it work. But generally, we would not expect *pyphe* to outperform scan-o-matics in terms of an easily measurable parameter like CV, since our changes address other aspects. We have expanded our analysis of wild strain growth curves using an additional 8 conditions as requested, and show details of the normalisation procedure for each (Figure 2—figure supplement 1).

B) The authors do not show the effect of normalization on the growth curves, which I think should be done, e.g. in Figure 2A. Moreover, growth curves should be shown on a log-scale such that the exponential phase can be readily distinguished.

Growth curves are not normalised, only the extracted maximum slopes are. We now show heatmaps of maximum slopes before correction, after grid correction and after additional rcmedian correction for 8 conditions in Figure 2—figure supplement 1. We do not show growth curves in Figure 2 on a log scale because they do not show cell numbers and do not follow an exponential pattern.

C) From a more general perspective: the whole of Figure 2 is based on a single, ideal environment (2% glucose YES) with canonical growth curves, but the conclusions are generalized to the method as such. I am not convinced that this is sufficient. I believe that it is fair to ask for expansion such that also a broad range of non-canonical growth curves are covered – preferentially from environments where growth is limited by other factors than the local glucose concentration. For comparison, Zackrisson et al., 2016 considered six different environments.

We have now expanded this dataset considerably using 8 additional and diverse conditions (Figure 2—figure supplement 1). These have been selected specifically to challenge our hypothesis and to result in as diverse growth dynamics as possible. We use rich media with mixed carbon sources and salt stress. We also use 4 different nitrogen sources of different quality (where glucose is not limiting for growth in poor nitrogen conditions). The correlation between endpoints and colony sizes is >0.9 for all with a median of 0.947.

5) I really, really appreciate the intentions of the authors in trying to extend their set-up to viability staining, which has the potential to count dead cells and resolve birth and death rates. However, I am not convinced that they yet have succeeded in their intentions.A) The authors do not really consider, or show, deaths over time and does not estimate death rates. Illuminating how death rates changes as a function of growth and environment, or really just highlighting how often it is substantially above zero at different parts of the growth curve, could advance the field substantially, as non-negligible death rates have confounding effects on key microbiological properties (e.g. Frenoy et al., 2018). A time resolved view on death would much improve the paper and we need to see it.

Thank you for raising this point. We have initially used colony redness as a readout for strain profiling, which can be obtained from the same plate with little extra investment. As such, we are more interested in how the readout can be used (e.g. as an input for clustering), which does not require mechanistic knowledge of how exactly this readout manifests itself. However, we agree that this is an interesting question and have now performed new timecourse experiments (new Figure 5). In summary, we find that redness scores are stable for at least 1 day after rapid growth has ended and that knock-out mutants with final redness are more red already at earlier points.

B) None of the method evaluation that is done for growth (Figure 2) is repeated for the viability staining. The reader has no real clue what precision and accuracy looks like over time, how errors are distributed across and within plates, in different environments, at different time points, how well the normalization works, how false positive rates compare to false negative rates etc. Since this is a method paper, I think this is an essential component.

Additionally to the timecourse experiment described above, we have added Figure 2—figure supplement 1 which describe within and between plate variation and normalisation strategies specifically for redness data.

C) From Figure 4D, E, it seems there is a huge variation in the registered colour intensity that is unrelated to whether cells are dead and alive. For much of the dynamic range, the registered redness seems to only reflect noise? And the fraction of live cells in a colony, in this span, has no real quantitative interpretation. The only reliable distinction seems to be the qualitative separation of colonies with many and few dead cells. This drastically reduces the usefulness of the method?

We respectfully disagree with this interpretation of the data. It is surprising to us that a simple scan of the bottom of a whole colony can detect viability to such high accuracy (r=0.88). This simple approach has possible confounding effects, such as distribution of dead cells within the colony (dead cells in thicker parts or on top are less easy to detect). Considering that other methods to assess viability of cells in a colony would usually require picking and resuspension of that colony and assessment by CFU counting, flow cytometry or microscopy, our approach is a game-changer which enables obtaining viability estimates at unprecedented throughput. We discuss caveats in the Discussion and are currently working on solutions to improve the sensitivity of this readout, using other imaging and quantification strategies. However, despite the not perfect correlation with the fraction of viable cells obtained by flow cytometry, colony viability scores show lower CVs and lower FUVs than colony sizes in our hand, making them a reliable and biologically meaningful readout for strain profiling and other functional genomics approaches.

“The only reliable distinction seems to be the qualitative separation of colonies with many and few dead cells.”

Both variables on the x- and y-axis cover the range of intermediate values so no strict binary grouping is apparent to us. We have further checked whether the data is compatible with the reviewer’s interpretation by dividing them into two groups (low and high viability based on FACS) and have computed the correlation to colony redness scores for both separately. Within both groups, the two readouts are still correlated (albeit to a lower extent; Author response image 5). We now mention this grouping in the main text.

**Author response image 5. sa2fig5:** Subgroup analysis of colony staining. We divided the data into two groups and computed the correlation separately. Both groups still show clear correlation (0.41 and -0.33) which is incompatible with the claim that the method allows a binary classification only. However, the within-group correlation is substantially lower than the overall correlation. Regardless, redness scores in themselves are highly reproducible and precise and therefore present an attractive fitness readout, the use of which does not require a detailed mechanistic understanding.

D) If I understand Figure 4C-E right, there seems to be a strong confounding effect of lysed cells when considering Pyphe colony redness: both lysed and alive cells reduce the redness. Hence, colonies with a high fraction of alive cells and colonies with a high fraction of lysed cells can show similar redness? In harsh environments, where a high fraction of cells are first killed and then lysed I imagine that results therefore will be very hard to interpret? This must be a serious shortcoming as compared to e.g. flow cytometry where lysed and alive cells seems to be well separated?

Indeed, in the flow cytometer, the redness is dead>alive>lysed, with three distinct populations visible. If this was similar in colonies, this would indeed be a problem. However, we propose that the redness in colonies is dead=lysed>alive. Dead cells stain bright red because phloxine B passively enters the cell and is not pumped out as it is in live cells. Lysed cells also cannot pump out the dye so they will be stained in the colony. When running samples on the flow cytometer, cells are resuspended in a buffer which quickly washes the dye out of the lysed cells. For these reasons, the colony redness largely reflects the number of dead and lysed cells. This conclusion is supported by the good correlation we observe between colony redness and the fraction of live cells [alive/(alive+lysed+dead)] by FACS. We also show in Author response image 6 that the correlation of colony redness to (alive+lysed)/(alive+lysed+dead) is weaker, suggesting that lysed cells do contribute to colony redness. We have not explained this issue more clearly in the manuscript text and added Figure 4—figure supplement 2.

**Author response image 6. sa2fig6:** The fraction of live cells (neither burst nor strongly stained in flow cytometer, left panel) better explains the colony redness score than the fraction of strongly stained cells only (right panel). This suggests that burst cells do contribute to staining in the colony (while being unstained in the flow cytometer). Note that the correlation breaks down for colonies with higher redness scores (which have more burst cells).

E) The authors general conclusion, that there is no overlap between the fraction of dead cells and colony growth, is conceptually very troubling. How can this not be the case, if they really capture population size growth and dead cells respectively? Surely, dead cells do not reproduce: growth, as a consequence, must slow? Or? For example, from Figure 3A, it seems that many colonies grow at a normal rate (i.e. reach an intermediate size), even though only 25% of cells are alive (i.e. the colony redness is 1.25). If, as stated, the detected death is completely disconnected from the detected population size growth, something is fundamentally strange with the detection.

We agree that these are interesting considerations. The simplest explanation is that growth and death are temporally uncoupled. While this does not seem to be the case for the knock-out mutants we investigated it might be the case in other scenarios, e.g. when working with wild strains. Similarly, they could be spatially decoupled. As not all cells in the colony are actively dividing, especially during later growth (Meunier and Choder, 1999) (and most likely in stress conditions), a subset of cells can die without overall colony growth being affected. This is supported by the uneven distribution of redness within the colony (which we currently do not capture with *pyphe*). Furthermore, colonies could sustain normal growth if viability is sacrificed for growth rate (akin to cells going into ‘overdrive’) (Nakaoka and Wakamoto, 2017). Which of these explanations is true will depend on the strains, conditions and incubation times and they can, of course, occur in combination. We have improved the Discussion based on the above points. We have several ongoing projects investigating these questions (with wild strains and mutants) and believe that *pyphe,* as it is currently presented, is well suited to give users the tools to explore these questions. Note that we have not calibrated redness scores to absolute viabilities. A redness score of 1.25 therefore does not mean that 25% of the cells are dead.

F) From Figure 3B it is clear that there often is a growth impact of phloxine B and that it depends heavily on the environment. One also wonders how genotype-dependent the impact of phloxine B on growth is? If phloxine B has a large impact on growth in many environments and on many genotypes – isn't there a serious risk for confounding effects when measuring both in parallel?

We do not agree with this interpretation of Figure 3B (and D). We do observe that otherwise identical conditions with and without phloxine cluster very closely together, meaning that the observed patterns in the data are clearly dominated by the condition (and not whether phloxine is used or not). Figure 3D further shows that the correlation between conditions with and without phloxine is consistently very strong (as strong as repeats of identical conditions in different batches). We already do point out the caveat that this does not prove that phloxine has no effect for few gene-condition pairs or for other conditions not tested.

Furthermore, one should consider that a general genotype-dependent impact of phloxine is conceptually not a problem. If phloxine is included in the assay condition and the control condition (as should be done), the genotype-dependent effect of phloxine will be normalised out. Conceptually, the addition of phloxine is just like adding/changing any other component in the media. This can and often will affect growth but is not a problem as long as the media used for the same experiment is the same.

We have now conducted an additional statistical analysis, testing for strain-condition pairs which have different fitness in the same condition with and without phloxine. We have an extremely high number of replicates for rich and minimal media control conditions with and without phloxine as they were repeated across many batches. We find one mutant, trehalose-6phosphate phosphatase Tpp1, which has lower fitness in media with phloxine than without. We now report this hit in the main text, also noting that the statistical non-rejection of the null hypothesis (that phloxine B has no effect) cannot be used as an indication that the null hypothesis is true.

Reviewer #3:The current manuscript describes a comprehensive pipeline that is a wrapper around already available tools and implements already described approaches (e.g. grid normalisation) and which can be used to analyse imaging data collected using flatbed scanners for high-throughput fitness screens. While the paper is very clear and well written and the code deposited in a public repository appears to be well crafted and documented, I am unsure there is enough novelty in this tool or in the experimental validation reported in the current manuscript to be of interest for the general readership of eLife. If I understand this correctly, there aren't any critical steps implemented in this pipeline which had not been reported or implemented before, which makes me think that a journal where this kind of tools are reported might be a better home for the current manuscript. Unfortunately I lack the expertise in the specific area of high-throughput phenotypic screens to be able to judge whether the substantial work presented here constitutes a technical improvement and a practical tool that might be widely used or rather a step change in the field. Without more competing arguments in favour of the latter and without a clear indication of a novel approach rather than implementation of already described tools and techniques I cannot fully support this manuscript for publication in eLife.

Many earlier sections in this response letter contain detailed comparisons of our strategy against others and several other points specifically deal with the question of novelty. We are confident that these make a convincing case for *pyphe*.

References:

Galardini M, Busby BP, Vieitez C, Dunham AS, Typas A, Beltrao P. 2019. The impact of the genetic background on gene deletion phenotypes in *Saccharomyces cerevisiae*. *Mol SystBiol* 15:e8831.

Ibstedt S, Stenberg S, Bagés S, Gjuvsland AB, Salinas F, Kourtchenko O, Samy JKA, Blomberg A, Omholt SW, Liti G, Beltran G, Warringer J. 2015. Concerted evolution of life stage performances signals recent selection on yeast nitrogen use. *Mol Biol Evol* 32:153–161.

[Editors' note: further revisions were suggested prior to acceptance, as described below.]

The manuscript has been improved but there are some remaining issues that need to be addressed before acceptance, as outlined below:All reviewers agree that you and your co-authors have responded adequately to the concerns that were raised and that the paper has matured significantly. That said, we would still recommend addressing two specific points in a bit more detail in the paper, so that the readers are at the very least made aware that these might be potential concerns.Firstly, we think it is important for growth measures to be as good proxies for population size as possible. Accounting for the non-linearity of optical and cell density is important, even if this is often ignored (because the difference is not huge, or because of technical issues).

We now state: "Image pixel darkness is known to scale non-linearly with true colony thickness/cell number (Zackrisson et al., 2016). Fitness estimates reported by *pyphe-analyse* are therefore related but not strictly the same as cell counts. If absolute population sizes are required for an experiment, the Scan-o-matic pipeline offers suitable calibration functionalities (Zackrisson et al., 2016)."

Second, there still is some concern about the lack of correlation between measured death and the measured fitness proxy – one reviewer is not sure that this is not due to one or both measures being afflicted by a large error. We understand that this is not easily solved, but believe it is fair to mention it explicitly in the paper as a potential concern.

In the Conclusions we now state: "Explaining the observed disparity between redness and size data should be a priority for future research and the explanation may depend on the strains, conditions, incubation times, or technical factors (or combinations thereof)."